# Quantum-inspired pedestrian mobility modeling: Applying probabilistic spatial simulation to urban walkability and thermal comfort in Sri Lanka

Malith Deshan[1], Amila Jayasinghe[2]*, Chethika Abenayake[2]

1 Urban Simulation Lab and Smart Mind UI and AI Research Group, Department of Town and Country Planning, University of Moratuwa, Moratuwa, Sri Lanka, 2 Department of Town and Country Planning, University of Moratuwa, Moratuwa, Sri Lanka

* amilabjayasinghe@gmail.com

## Abstract

Urban pedestrian movement is inherently uncertain, shaped by built form, microclimate, and time-varying crowding. This study develop a quantum-inspired probabilistic framework that models pedestrian presence as a spatial probability field derived from a composite potential integrating static structure (connectivity, crossings, barriers, visibility, points of interest) and dynamic drivers (crowd density, shade/thermal exposure). Applying the method to University Junction, Moratuwa, Sri Lanka, the study discretizes the domain to 5 m cells and 15-minute time bins, estimate factor weights via variational minimization, and solve an eigen-problem to obtain probability maps. The model reproduces diurnal reconfiguration of flows, concentrating midday probabilities in shaded, connected corridors and reducing presence on exposed verges. Link-level validation shows strong agreement with observed shares (Pearson's $r \approx 0.77$ evening; $\approx 0.71$–$0.77$ across periods) and realistic spatial autocorrelation. Scenario tests (temporary barriers, event footprints) re-equilibrate the probability field without retraining, revealing predictable re-routing to substitute corridors. Compared with a Boltzmann-type classical model and a space-syntax predictor, the proposed approach achieves higher fit and better spatial realism by explicitly encoding climate comfort and crowding. The framework yields policy-ready maps that support shade investment, crossing consolidation, and operational crowd management, providing an interpretable and transferable tool for assessing, managing, and allocating pedestrian space under uncertainty.

## Introduction

Human mobility is a core dimension of urban life and a fundamental concern of urban planning and design. Patterns of pedestrian movement influence accessibility, safety, social interaction, and the overall livability of cities and public space [1,2]. Understanding how people navigate urban environments is therefore essential for

**Data availability statement:** The author-generated code and derived, de-identified datasets used in this study are available at: GitHub repository: https://github.com/malithdeshan1222/PedestrianQuantumSimulation.

**Funding:** This study was supported by University of Moratuwa, Sri Lanka in the form of a grant awarded to AJ (Senate Research Committee Conference & Publishing Support Grant / UoM / 2026) and University of Moratuwa, Sri Lanka in the form of a salary for AJ. The specific roles of this author are articulated in the 'author contributions' section. The funders had no role in study design, data collection and analysis, decision to publish, or preparation of the manuscript.

**Competing interests:** The authors confirm no Competing Interests are to be declared.

designing urban space, transport systems, allocating resources, and creating equitable and sustainable public spaces [3,4]. As urban environments densify and become more dynamic—particularly in rapidly urbanizing, heat-exposed regions of the Global South—planners increasingly need tools that can anticipate how pedestrians respond to fine-scale changes in urban form, microclimate, and crowding, rather than only long-term network structure [5,6].

A wide range of modeling traditions has contributed to understanding pedestrian flows. Gravity and entropy-based models capture aggregate flows from generalized cost and mass, performing well at macro scales but often treating individual behavior and local context in a highly simplified way [7,8]. Space syntax and related network-centrality approaches emphasize the role of configurational accessibility and visibility, highlighting how street structure shapes "natural movement" [9,10]. Agent-based models and, more recently, deep learning frameworks can reproduce complex trajectories from data, but they are typically data-hungry, computationally intensive, and often opaque to planners [11,12]. Classical probabilistic formulations, such as Boltzmann-type or logit models, introduce randomness but still assume a fixed, energy-like cost for each path and usually treat alternatives as independent and static [6,13]. Collectively, these approaches struggle to represent two features that are central to contemporary street-level planning and designing: (i) non-deterministic, multi-route choice under uncertainty and (ii) the way microclimate, shade, and crowding reconfigure pedestrian patterns over the course of a day.

Recent work has begun to document how thermal comfort and microclimate systematically influence pedestrian behavior. Empirical and experimental studies show that pedestrians in hot climates adjust their routes to maximize shade, discounting sun-exposed distance and sometimes preferring longer shaded paths [14,15]. Urban street-canyon studies link street geometry, sky-view factor, and material properties to thermal indices such as PET (Physiological Equivalent Temperature) or UTCI (Universal Thermal Climate Index) with clear implications for outdoor comfort and walkability [16,17]. Daily and seasonal weather swings alter walking preferences; spontaneous events, demonstrations, and temporary barriers redirect flows; and climate-related discomfort shapes route choice and dwell time [18,19]. Microclimate mapping and walk-along surveys further demonstrate that local shade, radiation, and wind drive perceived comfort, walking experience, and sidewalk usage [14], [20]. However, these insights are often embedded in diagnostic or mapping exercises rather than in predictive mobility models. Many operational pedestrian models continue to treat climate as a background condition or post-hoc overlay, rather than as a first-order driver of route choice and presence in space. It indicated that urban planning and design increasingly needs tools that simulate behavior across a spectrum of possibilities, enabling planners to anticipate not only the most likely, but a distribution of plausible outcomes [21,22].

Such conditions reveal the inadequacy of purely deterministic models to capture uncertainty and temporal variation in pedestrian movement. In parallel, urban informatics has emerged as a convergent framework that integrates urban science, computer/data science, and geomatics to understand and shape cities using digital

traces, models, and decision-support tools [23,24]. Within this framework, researchers are exploring how advanced computation—AI, GeoAI, quantum and quantum-inspired methods—can help address complex mobility and design problems that are multi-scale, uncertain, and data constrained [24,25]. Quantum models of cognition and decision have shown that quantum probability can represent context-dependent, non-commutative, and "interference-like" effects in human decision-making more flexibly than classical probability [26,27]. In mobility research, quantum walks and quantum-inspired algorithms have been used to model super-diffusion and complex spreading patterns in urban human movement, highlighting their potential to capture multi-path, non-classical diffusion processes [28]. Yet explicit quantum-inspired formulations for pedestrian movement at the micro-scale of street design, with thermal comfort as a central driver, remain largely unexplored.

This paper responds to that gap. We introduce a quantum-inspired pedestrian mobility model that explicitly embeds uncertainty and microclimate into a spatial probability field, with the specific aim of supporting urban planning and design decisions. Conceptually, we treat pedestrians not as committing instantaneously to a single "shortest" path, but as occupying a superposed choice state over multiple feasible routes, each shaped by a composite potential that integrates built form, shade/thermal comfort, and time-varying crowding. Following ideas from quantum probability and quantum-inspired optimization, we represent the urban environment as an energy-like landscape and solve a Schrödinger-style eigenproblem to obtain a ground-state probability distribution over space and time. unlike conventional probabilistic models (e.g., logit or Boltzmann), which assign independent probabilities to discrete alternatives based solely on scalar utilities, our formulation couples neighboring cells through a Laplacian operator and enforces global coherence of the probability field. This allows the model to represent multi-route choice, spatial interference between nearby options, and smooth probability "channels" that correspond to emergent desire lines.

Within this framing, our study makes four contributions to urban planning and design and to the broader field of urban informatics. First, we develop a quantum-inspired, eigen-based pedestrian model that is explicitly grounded in urban planning and design variables—connectivity, crossings, barriers, visibility, points of interest—and in microclimate and crowding, rather than in abstract network states. Second, we systematically integrate thermal comfort and shade into the movement model, treating them as first-order components of the potential field and demonstrating how they reconfigure probability distributions across the day in a hot tropical context. Third, we benchmark the quantum-inspired model against a classical Boltzmann-type potential model and a space-syntax-based predictor, showing gains in statistical fit and, importantly, more realistic spatial patterns that planners can interpret and act upon. Fourth, we embed the method in an urban informatics workflow—using standard GIS layers, short, synchronized counts, and reproducible scripts—so that it can be transferred to data-constrained settings typical of many cities in the Global South.

Our empirical test study area is University Junction in Moratuwa, Sri Lanka: a compact, high-intensity node where strong pedestrian origins and destinations, discontinuous pedestrian infrastructure, and tropical heat combine to produce complex, time-varying mobility patterns. We discretize the area into 5 m grid cells and 15-minute time bins, construct static and dynamic potential fields, and estimate factor weights via a variational minimization procedure inspired by Variational Quantum Eigen solvers. The resulting probability maps are validated against observed link-level pedestrian shares and used to run planning-relevant scenarios (e.g., temporary barriers and event footprints).

In summary, we position this work as a novel urban planning and design application developed within the framework of urban informatics, drawing on quantum physics concepts to model pedestrian behavior under uncertainty in a way that is both mathematically rigorous and geographically interpretable. The remainder of the paper is organized as follows: Section 2 describes the study area and data; Section 3 details the quantum-inspired modeling framework and calibration; Section 4 presents results and comparative validation against classical baselines; Section 5 discusses implications for urban design and thermal-sensitive pedestrian planning, and Section 6 concludes with limitations and directions for future research.

 

## Study area and data

### Study area

The empirical setting is University Junction, Moratuwa, Sri Lanka (Fig 1), a compact, high-intensity node formed by the convergence of sub arterial road with local collector streets serving the University of Moratuwa and adjoining residential and commercial blocks. The area combines dense pedestrian generators (campus gates, bus stops, retail frontages, food courts, boarding houses) with frequent curbside activity (informal vending, ride-hailing, delivery). Sidewalk provision is discontinuous, crossings are largely unsignalized, and curb radii and vehicle turning volumes create recurrent pedestrian–vehicle conflicts. Tropical heat and limited shade impact walking comfort, especially at midday; activity peaks occur before morning classes, at lunch, and during late-afternoon egress. This blend of strong origins–destinations, constrained public-realm capacity, and climatic exposure make the junction a suitable test bed for probabilistic movement modeling under environmental and behavioral uncertainty.

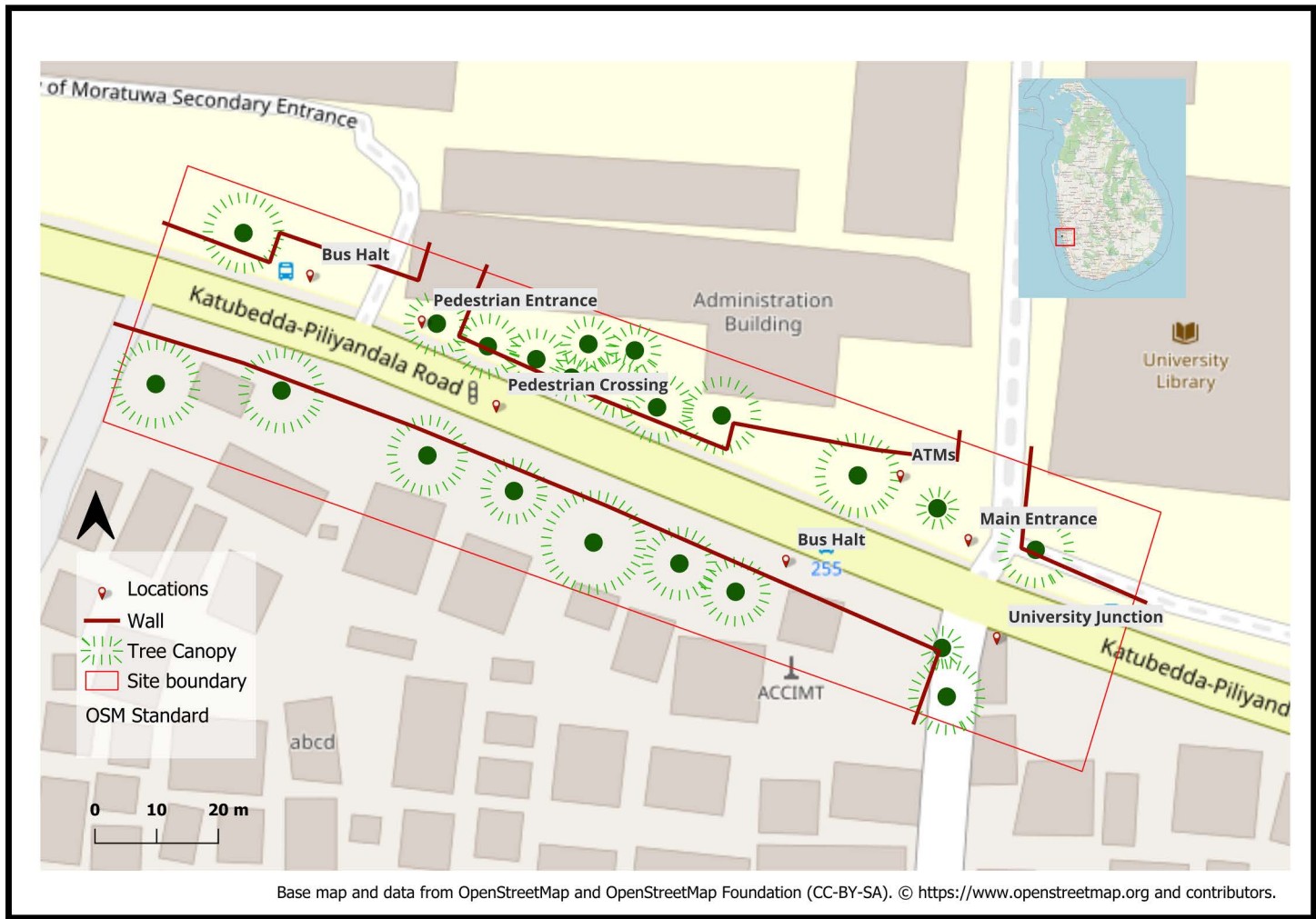

**Fig 1. Study Area showing site boundaries, walls, tree canopy, bus halts, pedestrian entrances, and crossings.** (Source: Prepared by authors, contains information from OpenStreetMap and OpenStreetMap Foundation, which is made available under the Open Database License).

For spatial analysis, we delineated a functional study boundary (~0.8 km²) centered on the junction and extending to adjacent blocks that contribute origin–destination flows (campus precincts, bus stops within ~500 m, and retail frontages on the sub atrial road). Street centerlines were simplified to a pedestrian network graph; mid-block passages and campus cut throughs were digitized to capture walkable permeability beyond vehicular links.

## Data collection and preparation

The input environment was first rasterized to a 5 m × 5 m base grid and a 15-minute temporal index, which aligns with the spatial scale of sidewalks and building frontages in the study area and with the counting protocol used for pedestrians and vehicles (Table 1). A 5 m cell size is commonly used in street-scale urban climate and thermal comfort studies as a compromise between resolving local shade and enclosure patterns and keeping computation and data density manageable [29,30]. This resolution is therefore fine enough to capture meaningful variation around crossings while avoiding severe sparsity in cells and counts. Similarly, 15-minute bins reflect standard practice in traffic and pedestrian monitoring for turning-movement and peak-period analysis, providing a balance between resolving intra-peak dynamics and maintaining sufficient observations per interval for reliable estimation [31,32],

To characterize temporal variation, the study conducted synchronized manual counts at fixed points during three daily periods representative of distinct demand and microclimatic conditions: Morning (07:30–09:30), Midday (12:00–14:00), and Evening (16:00–18:00). Counts were recorded at 15-minute intervals at two stations for pedestrians and two for vehicles, over three consecutive weekdays. The counting stations were located at the main gates, crossings, and conflict points of the junction and were mapped to the corresponding street segments and grid cells, so that observations represent link-level pedestrian presence within the modelled field. During each session, observers also captured photos and short video clips, which were later used to verify and refine the spatial allocation of pedestrians (e.g., which side of a link, which crossing arm) beyond what could be inferred from totals alone. This protocol ensured comparability across datasets and reduced temporal sampling bias; short gaps (≤15 min) were flagged for quality control and not interpolated unless both observers concurred. While individual trajectories were not collected, the quantum-inspired framework is calibrated to link-level presence and infers a continuous probability field within and between links; we identify the integration of GPS-/video-based trajectories as an important direction for future work (Table 2).

Other time-varying influences (e.g., retail spillover, signal timing) were documented but treated as static proxies (frontage type, legal crossings) or as scenario flags (temporary barriers/events), so that the limited dynamic dimensionality focuses on the two strongest behavioral drivers: crowding and shade

## Preprocessing and grid framework

All spatial layers were first prepared in QGIS before model integration.

**Table 1. Data sources.**

| Category | Factor | Metric/ Representation | Purpose in model | Source |
|---|---|---|---|---|
| Built form | Building density | Footprint coverage (ratio) | Enclosure/ edge effects | Open Street Map (OSM) |
| Built form | Building height | Stories/ height bands | Visibility & shadow casting | On-site survey |
| Vegetation | Tree height/ canopy | Crown footprint/ height class | Shade contribution | Field survey |
| Constraints | Walls/ barriers | Binary mask | Restricted zones | Field observations |
| Attractors | POI density | Kernel density/ counts | Destination pull | Field inventory, GIS digitizing |
| Movement structure | Integration (syntax) | Normalized integration | Movement effort/ accessibility proxy | DepthmapX |
| Visual field | Isovist openness | Local openness index | Perceived exposure/safety | Isovist app + GIS |
| Traffic | Vehicle density | Counts per 15-min bin | Ped–veh conflict context | Roadside survey |
| Pedestrian | Crowd density | Counts per 15-min bin | Time-varying load/ avoidance | Manual observations |
| Access | Ped. accessibility | Crossings, sidewalk continuity | Walkability checks | Field validation |

**Table 2. Dynamic variables retained.**

| Dynamic factor | How measured | Why included |
|---|---|---|
| Crowd density | 15-min pedestrian counts at fixed stations, mapped to links/grid cells using field sketches, photos, and short video clips | Captures avoidance/attraction and self-organization under space constraints at link level |
| Shade/ solar exposure | Shadow casting from buildings and trees derived from geometry and time-of-day; 15-min fields where available, otherwise period-specific means | Captures thermal comfort; major driver of route attractiveness in tropical street environments |

1. Grid generation and alignment:

The study discretized the study area with a regular 5 m × 5 m grid to support cell-based potentials and raster overlays (Fig 2). The grid provides a common index for merging continuous fields (e.g., shade index) and categorical masks (e.g., walls, POIs), and for snapping link-based measures to cells.

2. Rasterization of factors:

Vector layers (building footprints, walls, crossings, POIs, tree canopies) were converted to raster's at 5 m. Continuous surfaces (e.g., shade/solar) were computed per time slice and aggregated as needed (15-min bins to period means). DepthmapX-derived integration (and related centrality) was computed on the pedestrian network and transferred to the grid and segments via length-weighted overlays.

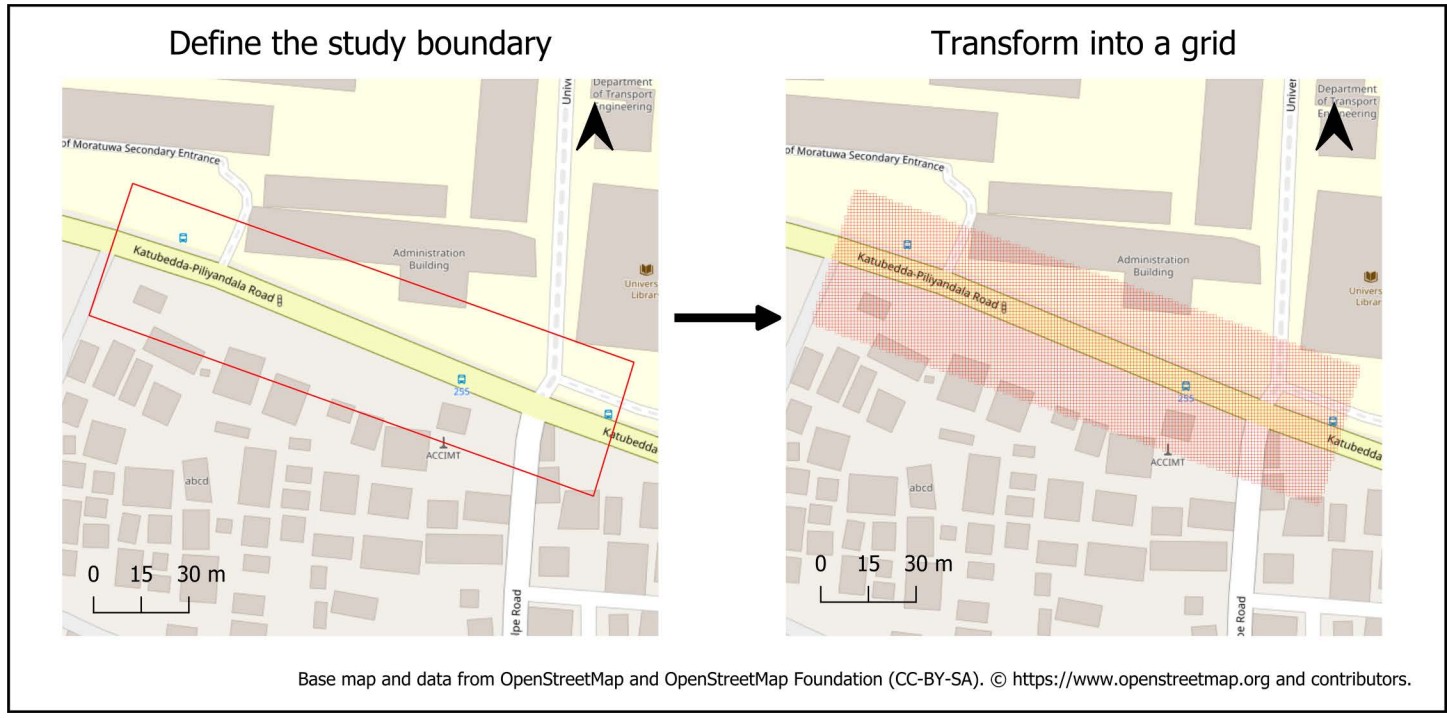

Base map and data from OpenStreetMap and OpenStreetMap Foundation (CC-BY-SA). © https://www.openstreetmap.org and contributors.

**Fig 2. Delineation of the University Junction study boundary and transformation to a 5 m × 5 m analysis grid (Source: Prepared by authors, contains information from OpenStreetMap and OpenStreetMap Foundation, which is made available under the Open Database License.**

3.  Standardization and masking:

Continuous predictors were z-scored; binary features were encoded 0/1. We applied a pedestrian-domain mask (side-walks/verges/plazas) to suppress non-walkable cells. To limit multicollinearity, highly correlated built-form indicators ($|r| > 0.8$) were screened, retaining the most interpretable representative (e.g., building edge/enclosure vs. raw footprint share).

4.  Temporal indexing:

Dynamic layers were indexed at 15-minute resolution to match observations. When only period measures were available (e.g., midday shade), values were broadcast to constituent bins. Pedestrian and vehicle counts were stored both as absolute volumes and shares of the daily total to facilitate cross-day comparison and scale-free validation.

5. Quality control.

We performed cross-checks between field sketches and OSM geometry, verified crossing locations and barrier continuity on site, and reconciled POI records with frontage observations. Any discrepancies were resolved by on-site photographs and timestamped notes.

## Methods and computational framework

This study proposed a quantum-inspired probabilistic framework to model pedestrian presence as a spatial probability distribution that varies with built form, microclimate, and crowding. The approach is "quantum-inspired" in the modeling sense—the study borrows the mathematics of superposition, potentials, and eigen-solutions to represent non-deterministic behavior; the study do not claim physical quantum effects in urban systems. The method is designed to serve the aims of urban informatics and planning: it translates complex urban behavior into geographically interpretable fields and decision-

### Conceptual framework

Pedestrian movement is treated as non-deterministic: individuals simultaneously consider multiple routes, evaluate competing attractions/repulsions (e.g., shade vs. distance), and then commit to a choice. The study represents this using two core ideas:

Superposition (choice set): before committing, a pedestrian is represented by a probability amplitude over feasible locations (cells or links).

Potential landscape (context): the urban environment is encoded as a spatial potential field in which low values denote attractive, low-effort, climate-comfortable conditions, and high values denote barriers, heat exposure, or conflict with traffic.

At decision points, the distribution "collapses" to realized paths; at the aggregate level the study observe probability of presence in space and time (Fig 3).

In this framework, core ideas are translated into planning-relevant interpretations as indicated in Table 3.

**Urban potential landscape.** The study models the urban context as a composite potential:

$$V(\boldsymbol{x}, t) = V_{\text{urbs}}(\boldsymbol{x}) + V_{\text{civitas}}(\boldsymbol{x}, t),$$

where $\boldsymbol{V_{\text{urbs}}}(\boldsymbol{x})$ summarizes static spatial structure (built edge, integration/centrality, legal crossings, barriers, visibility openness, POIs), and $\boldsymbol{V_{\text{civitas}}}(\boldsymbol{x}, t)$ represents dynamic drivers (crowd density, shade/solar exposure, scenario flags). Each component is a weighted sum of standardized inputs:

$$V_{\text{urbs}}(\boldsymbol{x}) = \sum_i \alpha_i F_i(\boldsymbol{x}), V_{\text{civitas}}(\boldsymbol{x}, t) = \beta_1 P_d(\boldsymbol{x}, t) + \beta_2 S_a(\boldsymbol{x}, t),$$

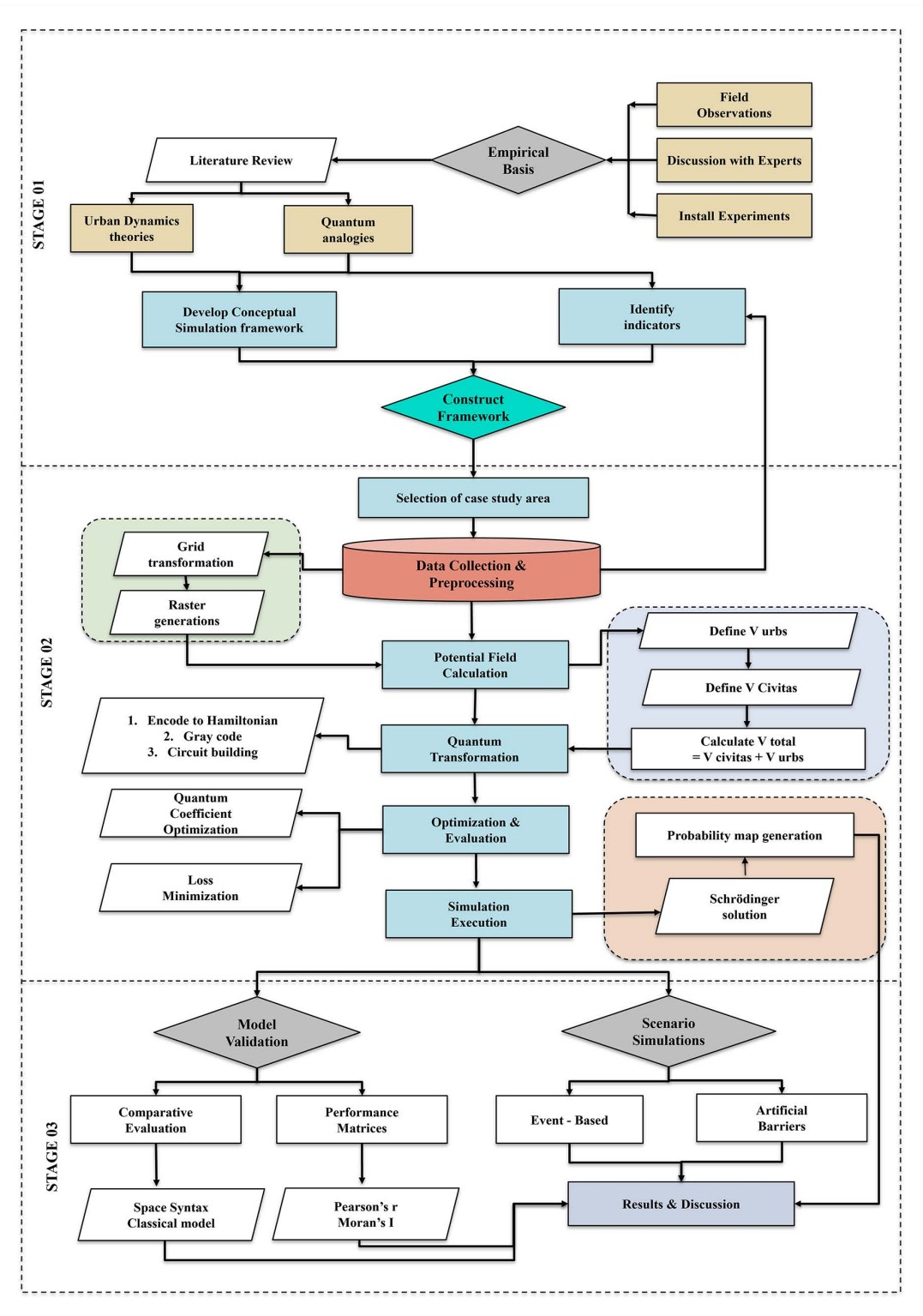

**Fig 3. Workflow of the quantum-inspired pedestrian modelling and validation framework (Source: Prepared by authors).**

**Table 3. Mapping of concepts to planning.**

| Quantum-inspired concept | Urban planning interpretation | Example at University Junction |
|---|---|---|
| Superposition | Concurrent evaluation of multiple routes | Shaded arcades vs. shorter sunlit path |
| Potential field | Spatial effort/comfort surface | Midday heat as repulsive; crossings as attractive |
| Coupling/interaction | Local conditions influence nearby flows | Queue spillover reduces adjacent curb appeal |
| Collapse (decision) | Route commitment at junctions | Peak-hour gate preference |

with $\alpha_i, \beta_k$ denoting learned weights and $F_i$ the static factors (e.g., integration, barriers). $P_d$ is crowding (15-min share) and $S_a$ is a shade/thermal attractiveness index.

Interpretation is geographical and policy-relevant: low $V$ highlights walkable, shaded, connected micro-corridors likely to attract pedestrians; high $V$ flags repulsive areas (exposure, conflict, obstruction) where interventions may be needed as indicated in Table 4.

**Spatial and temporal encoding.** The discretize the study area to a 5 m × 5 m grid aligned with a 15-minute temporal index. Static layers are rasterized; dynamic layers are computed per bin (or broadcast when only period means are available). All continuous predictors are z-scored; binary features are 0/1. To reduce redundancy, we screen collinearity and keep the most interpretable indicator within correlated groups (e.g., enclosure vs. raw footprint share).

For computational stability and interpretability, the study maintains a pedestrian-domain mask (sidewalks/verges/plazas) and map link-based measures (e.g., integration) to cells by length-weighted overlay. The grid acts as a shared spatial index that supports reproducibility across sites.

**Optimization and eigen-solution.** Given $V(x, t)$, the study seek a probability field $P(x, t)$ describing where pedestrians are most likely to be. The study use the analogy of a ground-state solution in a potential well. Let $H(V)$ denote a Hamiltonian operator constructed from the discretized Laplacian (spatial smoothness) and the potential $V$. The study solve the variational eigen-problem:

$$H(V)\,\psi \;=\; E\,\psi, \quad P(x, t) \;=\; |\,\psi(x, t)\,|^2,$$

where $\psi$ is the probability amplitude and $E$ the minimal expected "effort." Intuitively, probability mass concentrates in low-potential corridors while remaining spatially coherent due to the Laplacian term (discouraging isolated spikes).

Weights $\{\alpha_i, \beta_k\}$ are estimated by variational minimization of an objective that balances fit to observations and spatial regularity:

**Table 4. Inputs composing $V_{\text{urbs}}$ and $V_{\text{civitas}}$ with signs (±), units, and hypothesized effects.**

| Component | Factor | Sign | Units/ Scale | Hypothesized effect |
|---|---|---|---|---|
| $V_{\text{urbs}}(x)$ | Building edge/ enclosure | − | m per cell (z) | Strong edges/legibility reduce effort |
| | Integration (syntax) | − | index (z) | Higher accessibility attracts flows |
| | Legal crossing proximity | − | distance (inv./z) | Easier/safer crossing lowers effort |
| | Barriers/ walls | + | 0/1 | Physical constraints repel movement |
| | Visibility openness (isovist) | − | index (z) | Openness/readability attracts |
| | POI density/ active frontage | − | counts/ kernel (z) | Destinations pull pedestrians |
| $V_{\text{civitas}}(x, t)$ | Crowd density | + | share per 15 min (z) | Local crowding induces avoidance |
| | Shade/ thermal comfort | − | index 0–1 (z) | Shade increases comfort/attraction |
| | Event/ barrier flag | + | 0/1 | Temporary occupation repels |
| | Transit/class pulse | − | 0/1 | Pulses attract gates/stops |

$$\min_{\alpha,\beta}\ \mathcal{L}(\alpha,\beta)\ =\ \underbrace{1-\rho,(P_{\text{pred}},P_{\text{obs}})}_{\text{lack of fit}}\ +\ \lambda\underbrace{\|\nabla P_{\text{pred}}\|_2^2}_{\text{smoothness}},$$

where $\rho$ is a correlation metric computed on link-aggregated or cell-based probabilities and $\lambda$ controls smoothness. The minimization is implemented via a hybrid variational routine (quantum-inspired VQE style) that iteratively updates $\{\alpha,\beta\}$, rebuilds $V$, and recomputes $\psi$.

The solution is a geographically interpretable probability surface sensitive to microclimate and configuration, not a black box forecast. It yields actionable maps for reallocating shade, widening paths, or managing crowding.

**From amplitudes to pedestrian probability maps.** For each time bin $t$:

1. Build $V(\cdot,t)$ from current weights and inputs.

2. Solve $H(V)\psi = E\psi$ (ground-state approximation).

3. Compute $P(\cdot,t) = |\psi|^2$ and normalize to the pedestrian domain.

4. Apply a light Gaussian smoothing ($\sigma$ chosen via cross-validation) to suppress numerical noise while preserving corridors.

5. Aggregate to links (if needed) by length-weighted averaging to compare with observed counts.

Outputs are probability heatmaps whose cold–hot gradient directly reflects relative attractiveness/repulsiveness. Because $P$ is bounded $[0,1]$ and normalized within the domain, cross-day comparisons are robust to total volume differences.

## Calibration, validation, and baselines

The study calibrates on odd hour bins and validates on even hour bins to reduce temporal leakage. The study evaluates at two levels:

1. Cell-level calibration: rank correlation between predicted and observed presence (where observations support cell inference).

2. Link-level validation: Pearson/Spearman between predicted probability shares (aggregated to links) and observed shares. We also report Moran's I on residuals to detect unmodeled spatial structure.

Baselines for comparison

1. Classical probabilistic model (Boltzmann-type): probability $\propto \exp(-\text{scaled cost})$, using the same inputs but without eigen-regularization.

2. Space Syntax baseline: predicts flows from integration/visibility alone (Depthmap-derived), rescaled to probability shares.

Performance reporting. We present correlation metrics, calibration plots, and Bland–Altman style residual maps to identify systematic biases (e.g., underprediction near ad-hoc crossings).

**Sensitivity, parsimony, and scenarios.** Sensitivity to inputs. The study assesses the contribution of each factor by re-estimating with leave-one-out inputs and reporting the drop in validation $\rho$. The study also perturbs weights by ±1 SD to show how $P$ reconfigures (local elasticities). This identifies high-leverage levers for planning (e.g., adding shade vs. adding a crossing).

Parsimony. To avoid overfitting, the study restricts dynamic factors to the two most influential temporal drivers in this climate and context—crowd density and shade—while incorporating other influences through static proxies or scenario flags.

Scenarios. The study encodes two policy-relevant interventions:

1. Barrier closure of one sidewalk segment.

2. Event occupation of a plaza cell set.

In each case we recompute $V$ and $P$ to visualize re-routing and identify pressure points for temporary management.

**Reproducibility and implementation details.** Workflows are fully reproducible from raw data to maps. Spatial preprocessing used QGIS (digitizing, rasterization) and DepthmapX (integration/visibility). Modeling/validation ran in Python (NumPy/SciPy/scikit-learn; sparse linear algebra). Layers were processed in a meter-based projected CRS and exported in WGS84.

The computational domain is a 5 m grid over the pedestrian envelope; dynamic inputs (crowding, shade) use 15-minute bins. Continuous predictors are z-scored; binaries are 0/1; a pedestrian mask excludes non-walkable cells. Collinearity ($|r| > 0.8$) is screened to keep a single, interpretable indicator per group.

The potential field $V(x, t) = V_{urbs}(x) + V_{civitas}(x, t)$ combines static (built edge, crossings, barriers, POIs, integration, openness) and dynamic (crowding, shade) layers with learned weights $\{\alpha_i, \beta_k\}$. We solve the variational eigen-problem $H(V)\psi = E\psi$ and obtain pedestrian presence $P = |\psi|^2$, normalized within the pedestrian domain. Training minimizes (1 – correlation) to observations plus a Laplacian smoothness term; optimization uses gradient updates with early stopping and weight clipping. A light Gaussian smoothing (σ via cross-validation) reduces numerical speckle.

Validation: cell-level rank agreement and link-level Pearson/Spearman on probability shares, plus Moran's I of residuals. Baselines: Boltzmann-type model (same inputs, no eigen-regularization) and Space Syntax predictor. Outputs are open formats: GeoTIFF (float32) for $V$ and $P$; GeoPackage/GeoJSON for vectors; CSV/Parquet for link time series. A YAML config records CRS, parameters (σ, $\lambda$, seeds), paths, and checksums; a single runner script rebuilds the stack on any workstation (≥ 16 GB RAM).

In sum, the proposed quantum-inspired framework recasts pedestrian movement as a probability field shaped by a composite, geographically interpretable potential that integrates built form, shade/thermal comfort, and time-varying crowding. By estimating weights via a variational eigen-solution and validating against observed link shares, the model balances explanatory transparency with robust spatial regularity. Relative to deterministic overlays and single-metric predictors, it captures uncertainty and multi-route choice while remaining operational at 5 m and 15-minute resolutions. The following section reports calibration and validation results, compares performance to classical and space-syntax baselines, and examines sensitivity to inputs and policy-relevant scenarios.

## Results and comparative validation

This section reports the performance of the quantum-inspired pedestrian model at University Junction, emphasizing findings that are actionably useful for urban planning and design—i.e., understanding where, when, and why pedestrians concentrate, and how those patterns respond to microclimate and temporary disruptions. Results are organized from parameter learning, field interpretation, probability maps, scenario response, comparative validation and planning implications.

## Coefficient learning and convergence

The hybrid variational routine converged within ~50 iterations, with the objective (1 – correlation + smoothness penalty) decreasing monotonically. Learned weights assign greatest influence to dynamic drivers, confirming the study's premise that temporal and comfort factors govern micro-scale routing in a tropical setting (Table 5).

• Dynamic weights: Crowd density (C1 = 0.5869) and Shade attraction (C2 = 0.4131) dominate the civitas term.

**Table 5. Optimized urbs (U1–U8) and civitas (C1–C2) coefficients.**

| Index | Factor | Value |
|---|---|---|
| U1 | Building Density | 0.138852 |
| U2 | Building Height | 0.229278 |
| U3 | Shadow Intensity | 0.127974 |
| U4 | POI Density | 0.090927 |
| U5 | Wall Constraint | 0.15178 |
| U6 | Integration | 0.121113 |
| U7 | Isovist | 0.114626 |
| U8 | Vehicle Density | 0.02545 |
| C1 | Crowd Density | 0.586935 |
| C2 | Shadow Attraction | 0.413065 |

**Note:** Values are reported after proxy-based multicollinearity screening (S1-S2 Tables); C1 and C2 are the two dynamic weights.

- Static weights: Building height (U2 = 0.2293) and Building density (U1 = 0.1389) are the largest contributors within the urbs component; walls/barriers (U5 = 0.1518) and integration (U6 = 0.1211) also carry meaningful signal. Vehicle density (U8 = 0.0255) is weak, reflecting limited pedestrian–vehicle substitution on the observed links.

Before interpreting these coefficients as independent effects, we examined interdependence among the spatial covariates using Pearson/Spearman correlation matrices and variance inflation factors (VIFs) (S1–S2 Tables in supplementary materials). Built-form indicators such as building density, building height and shadow intensity are strongly coupled, and movement-related variables (pedestrian density, POI density, road accessibility, street centrality and vehicle density) form another collinear group, reflecting their shared dependence on the same street-canyon and network geometry. Rather than assuming full independence, we therefore treat the estimated coefficients as acting on a smaller set of latent constructs: U1 and U2 together proxy urban form intensity and enclosure, U3 proxies microclimatic/ thermal comfort, U5 and U8 approximate conflict and friction, and U6 and U7 capture configuration and local visibility. In other words, the weights in S2 Table should be read as group-level signals for these broader constructions, not as isolated, fully separable physical dimensions.

Interpretation. Crowding and shade shape route choice throughout the day; static configuration remains important but secondary. Height and edge density matter because they co-determine enclosure (cooler, visually legible edges) and shadow casting, which the model learns to reward indirectly through lower potential. The dominance of C1 and C2 confirms that, in this tropical junction, pedestrian presence is most sensitive to dynamic crowding and shade conditions, with built form providing the structural scaffold within which these effects play out (Fig 4).

## Interpreting the potential landscape

The study decomposes the learned surface into structural $V_{urbs}$ and behavioral $V_{civitas}$ fields to explain where low-effort corridors emerge (Fig 5).

- $V_{urbs}$ (static): high values align with barriers, unshaded exposed verges, and low-connectivity fragments; low values trace legal crossings, connected edges, and visually open arcades.

- $V_{civitas}$ (dynamic): at midday, shade-rich segments drop in potential (more attractive), whereas exposed segments rise; during peaks, local crowding slightly raises potential on already dense links (soft avoidance), redistributing probability to nearby substitutes.

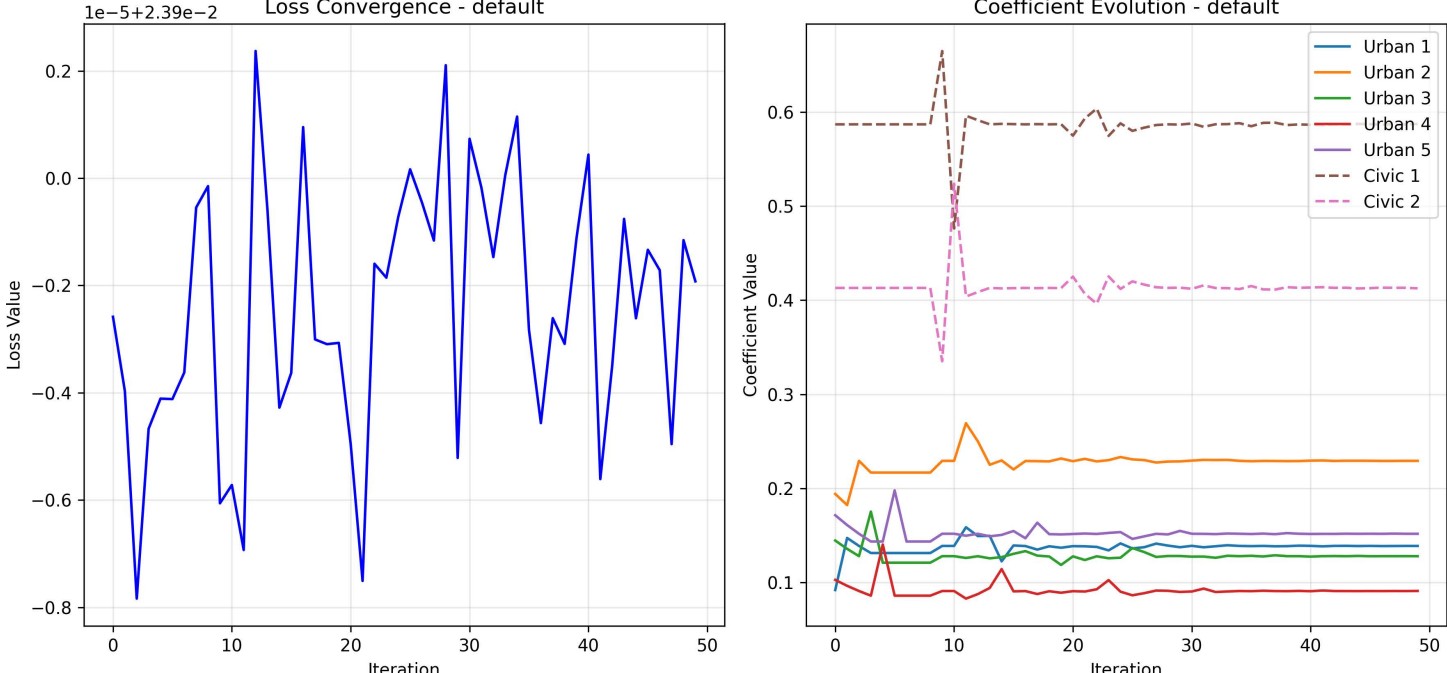

**Fig 4. Loss convergence and coefficient-trajectory plots (iterations vs. weight), (Source: Prepared by authors).**

Total potential $V(x, t) = V_{urbs} + V_{civitas}$ therefore deepens shaded, connected corridors and raises (repels) hot, congested, or constrained cells. In the Schrödinger-like formulation, probability mass concentrates in low potential regions, yielding coherent "probability channels" that match observed desire lines.

under the Open Database License) Note: Morning, midday, and evening maps of the static structural potential $V_{urbs}$ (top row), the dynamic behavioral potential $V_{civitas}$ (middle row), and the total potential $V_{total} = V_{urbs} + V_{civitas}$ (bottom row). The red outline shows the modelled junction grid over the local street and building layout. All panels share the same color scale, where blue indicates low potential (more attractive/ lower walking effort) and red indicates high potential (more repulsive/ higher walking effort).

## Probabilistic pedestrian distributions

Squaring the eigen-solution amplitude yields normalized probability fields $P(\cdot, t) = |\psi|^2$. The model reproduces the diurnal reconfiguration of pedestrian presence (Fig 6):

- Morning (07:30–09:30). Highest probabilities along the central access spine, near the main gate and bus-halt side. Probability tails extend to secondary gates as classes begin; localized clusters appear where early shade persists.

- Midday (12:00–14:00). Probability contracts into shaded arcades and tree-lined segments, with reduced presence on exposed verges. Broader but lower-intensity spread near canopied frontages indicates lingering/loitering comfort.

- Evening (16:00–18:00). Clustering strengthens near eastern segments, exits, and POI-adjacent links, consistent with homebound flows and cooler conditions. Probability channels widen where both connectivity and shade improve.

Aggregating $P$ to links produces probability shares that align closely with observed counts at the same temporal bins. Probability calibration is scale-free, aiding cross-day comparisons.

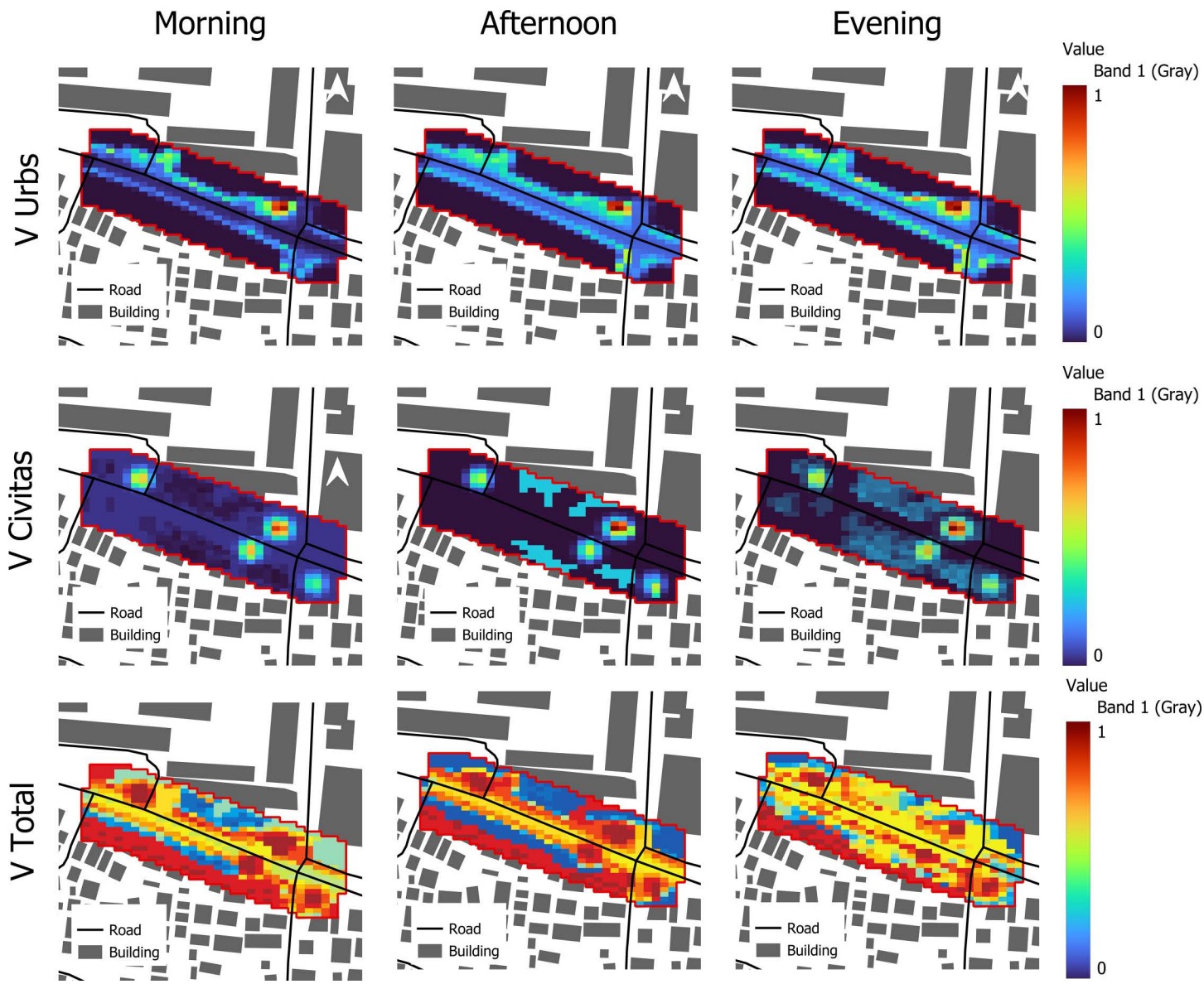

**Fig 5. Static, dynamic, and total potential fields by time of day.** (Source: Prepared by authors, Contains information from OpenStreetMap and OpenStreetMap Foundation, which is made available under the Open Database License) Note: Morning, midday, and evening maps of the static structural potential $V_{urbs}$ (top row), the dynamic behavioral potential $V_{civitas}$ (middle row), and the total potential $V_{total} = V_{urbs} + V_{civitas}$ (bottom row). The red outline shows the modelled junction grid over the local street and building layout. All panels share the same color scale, where blue indicates low potential (more attractive / lower walking effort) and red indicates high potential (more repulsive / higher walking effort).

To examine these patterns at a course, planning-relevant scale, the study area was divided into nine spatial zones reflecting functional sub-areas of the junction (Fig 7), including the main gate, bus halt frontages, side entrances, and the eastern T-junction. Aggregating link-level probabilities to these zones shows that pedestrian density varies markedly over the day (Fig 8). Morning flows are moderate and relatively evenly distributed across zones as students and staff arrive. At midday, probability contracts toward shaded and more central zones, consistent with thermal discomfort on exposed verges. In the evening period, zone-level probabilities peak, with the highest concentrations in Zones 6, 7 and 8, which

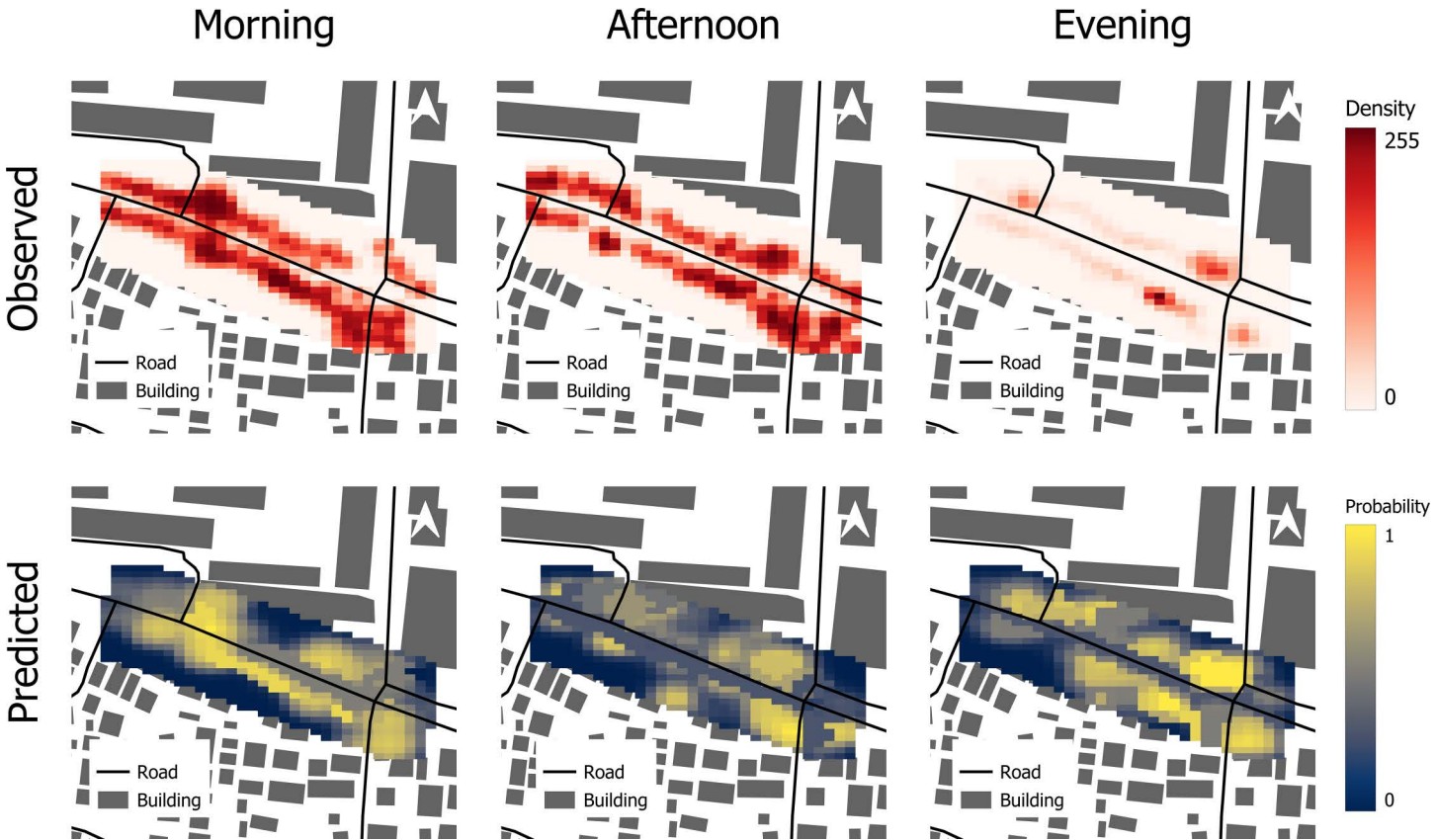

**Fig 6. Side-by-side observed vs. predicted maps per period; include difference (pred—obs) inset.** (Source: Prepared by authors, contains information from OpenStreetMap and OpenStreetMap Foundation, which is made available under the Open Database License).

correspond to the critical pedestrian corridors around the bus halt, main entrance, and the eastern T-junction with nearby POIs.

## Scenario responsiveness

Two stress tests evaluate planning-relevant adaptability without retraining, by modifying $V$ to reflect temporary conditions (Fig 9).

1. Parking barrier + a typical gate opening (event day). Raising potential along the parked edge and lowering it at the exceptionally open main gate (afternoon) deflects probability away from the obstructed verge and pulls it toward the gate and adjacent shaded approaches. The re-equilibrated $P$ matches field notes on event-day crowding.

2. Protest/repulsion at the main gate. Elevating potential within the protest footprint bypasses this zone and reinforces alternative corridors, especially the shaded offset route. The probability field adapts instantaneously via the eigen-solution—no parameter refit required.

   Implication. The framework can prototype operations (temporary barriers, event management, Open Streets) in minutes by editing scenario layers and recomputing $P$.

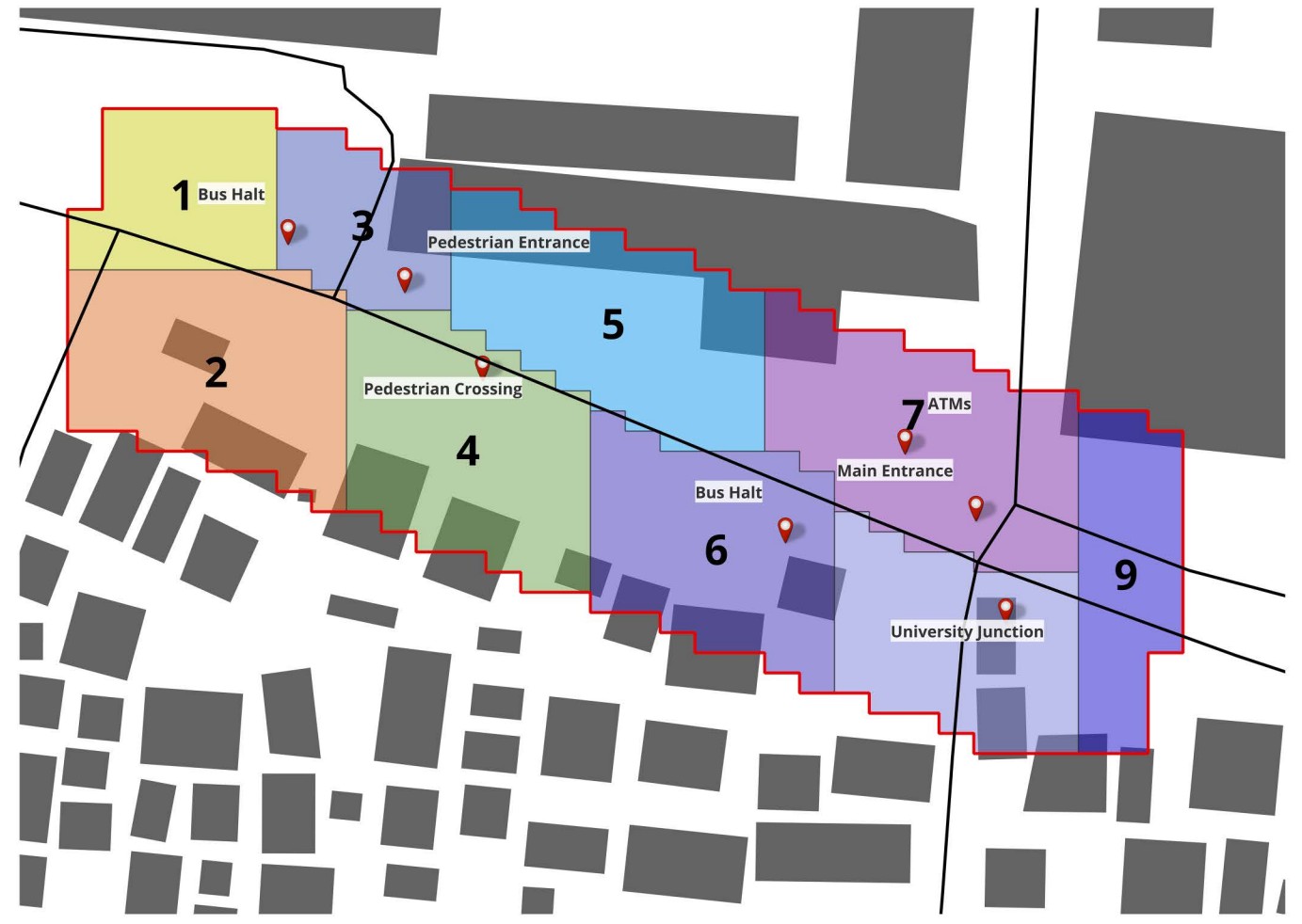

**Fig 7. Zoning of the study area (Source: Prepared by authors, contains information from OpenStreetMap and OpenStreetMap Foundation, which is made available under the Open Database License).**

## Comparative evaluation against baselines

We benchmark the quantum-inspired model against (i) a Boltzmann-type classical potential model using the same inputs and (ii) a Space Syntax predictor based on integration/visibility only as indicated in Table 6 and Fig 10.

- Quantum-inspired: Highest alignment with observed evening behavior ($r \approx 0.767$; $\rho \approx 0.727$), and Moran's $I \sim 0.75$, close to the observed 0.78. This indicates that the model reproduces both the strength and the spatial patterning of link-level pedestrian presence.

- Classical (Boltzmann-type): Reasonable rank fit ($\rho \sim 0.62$) but lower linear agreement ($r \sim 0.41$); high Moran's $I$ ($\sim 0.89$) indicate pronounced over-clustering, i.e., excessive concentration in a few corridors. Space Syntax: Weak agreement ($r \sim 0.14$; $\rho \sim 0.22$; $I \sim 0.61$), reflecting the omission of microclimate and dynamic behavior—it captures configuration but not climate-comfort or crowd avoidance. because the quantum-inspired model explicitly includes a spatial regularization (smoothness) term, we interpret Moran's $I$ comparatively rather than as a standalone proof of behavioral realism:

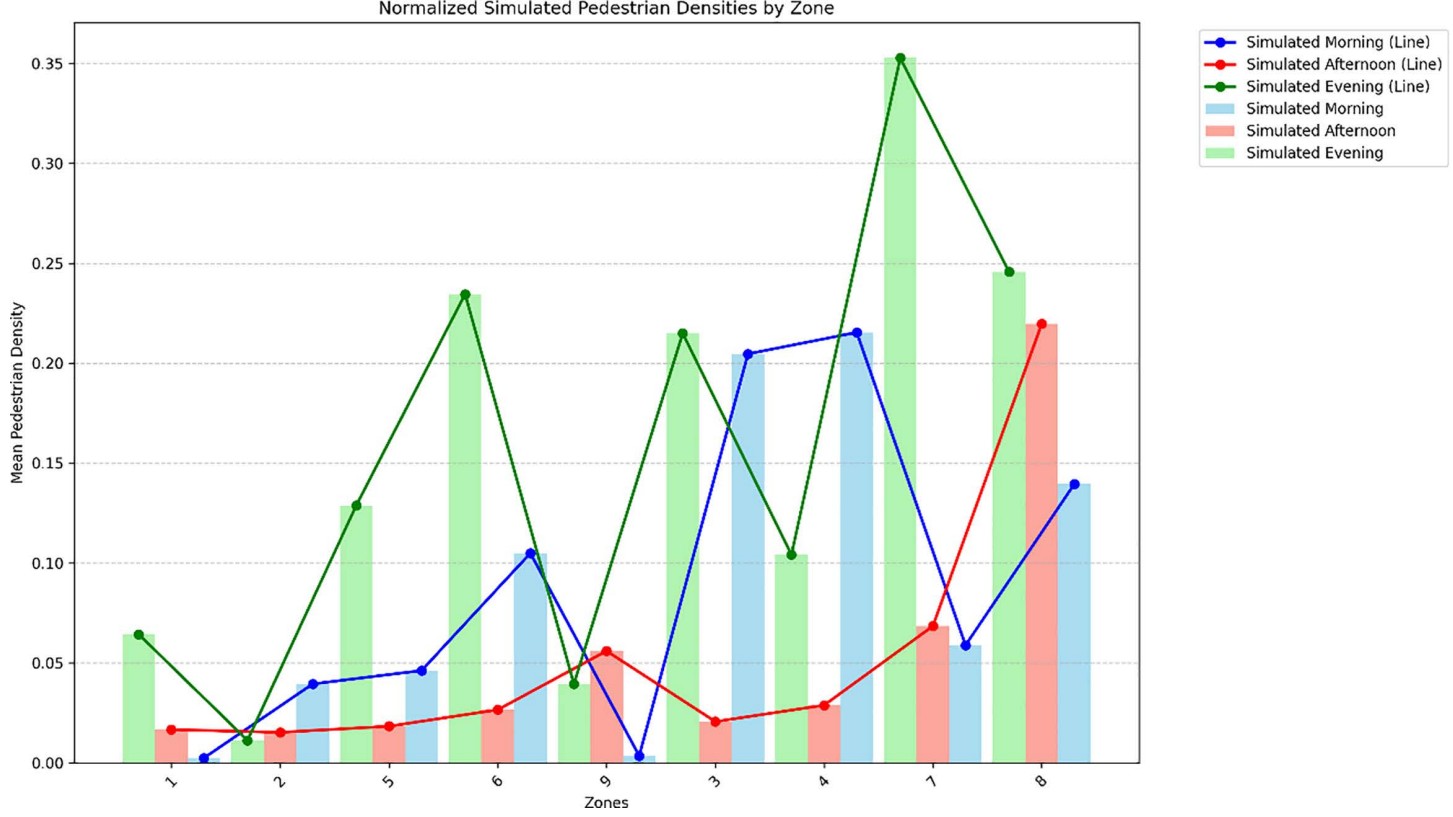

**Fig 8. Zone-level probability shares by period (Source: Prepared by authors).**

smoothing will, by design, enhance spatial autocorrelation to some extent. In this context, Moran's I is used as a consistency check that the predicted clustering is of a similar magnitude to the observed pattern, while the primary evidence of predictive skill comes from Pearson's r and Spearman's ρ.

The quantum-inspired surface integrates configuration *and* thermal/crowding signals, producing more realistic, temporally adaptive probability maps useful for climate-sensitive street design, shade prioritization, and operational crowd management. The comparative results show that this integration improves value correspondence (r, ρ) while avoiding the excessive over-clustering of the classical potential and the under-structuring of the space syntax baseline.

### Stability, uncertainty, and quantum-inspired behaviour temporal stability across periods

The period-wise evaluation shows that the model's predictive performance is consistently strong across different times of day while still reflecting meaningful temporal variation (Table 7). Pearson's r ranges from 0.71 at midday (r = 0.709) to approximately 0.77 in the morning and evening (r ≈ 0.766–0.767), indicating that the model captures observed pedestrian patterns most effectively during peak activity periods. Spearman's ρ follows a similar pattern and remains close to Pearson's r in all three periods (ρ ≈ 0.690–0.748), implying that even when absolute prediction errors increase, the rank ordering of high- and low-usage links is preserved. RMSE is highest at midday and lowest in the evening, further confirming that accuracy is temporally dependent but remains within a stable and interpretable range. Overall, performance is not dominated by a single favorable time slice.

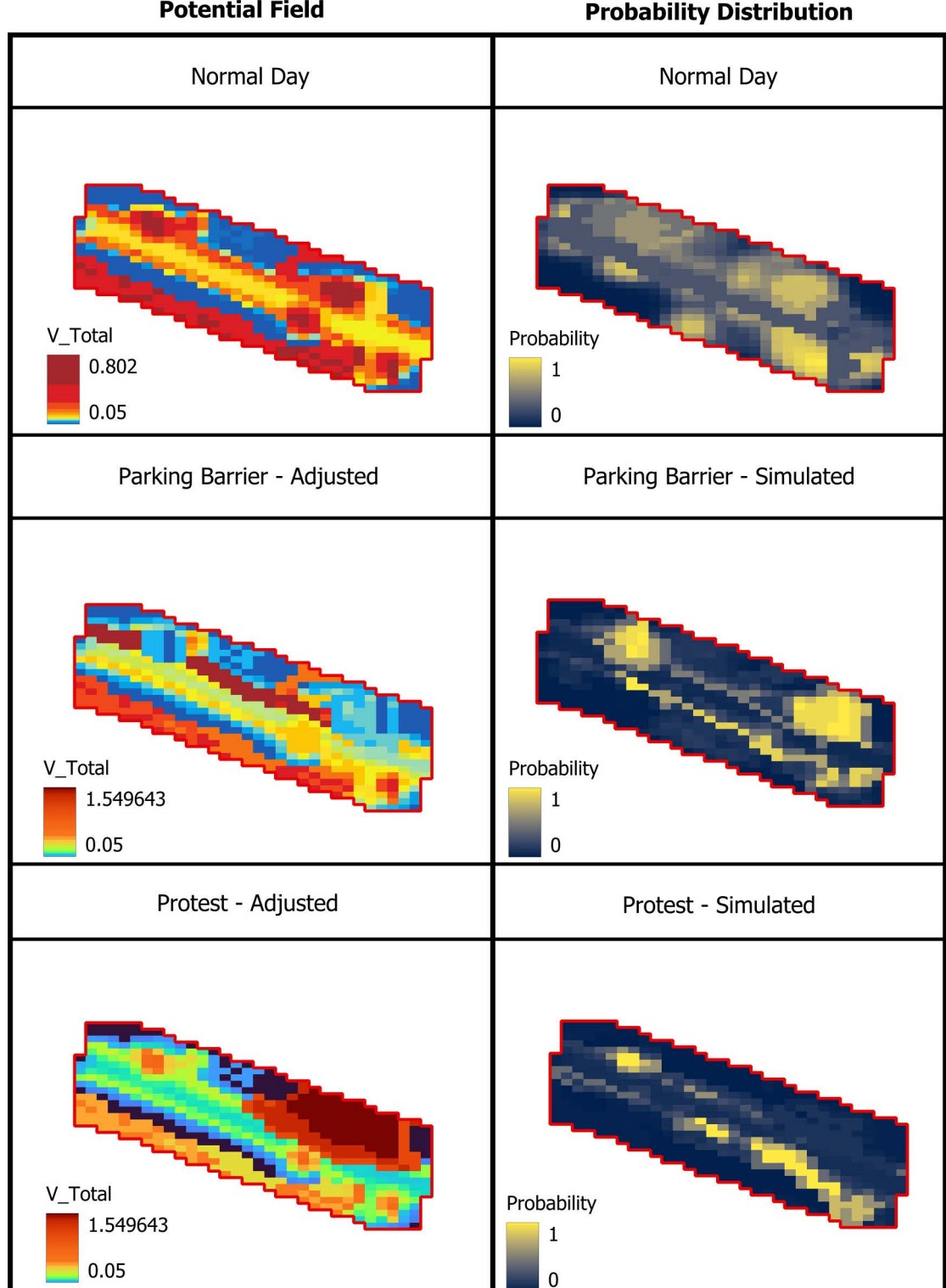

**Fig 9. Before/after V and P for each scenario, with arrows showing major re-routes (Source: Prepared by authors, contains information from OpenStreetMap and OpenStreetMap Foundation, which is made available under the Open Database License).**

**Table 6. Side-by-side metrics for the quantum-inspired model and two baselines (classical Boltzmann-type potential and space syntax predictor).**

| Model | Spatial Agreement (Moran's I) | Relationship (Pearson's r) | Relationship (Spearman's Rank) |
|---|---|---|---|
| Quantum-Inspired | 0.7787 | 0.7669 | 0.7369 |
| Classical Model (Boltzmann curve) | 0.8895 | 0.4117 | 0.5521 |
| Space syntax | 0.6109 | 0.1419 | 0.2223 |
| Observed | 0.7843 | – | – |

Note: Moran's I is reported as an indicator of relative spatial clustering and is interpreted in light of the explicit smoothness term in the quantum-inspired formulation, rather than as a standalone proof of behavioral realism.

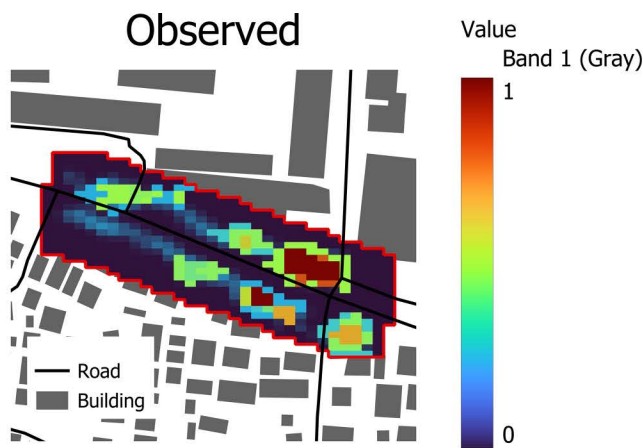

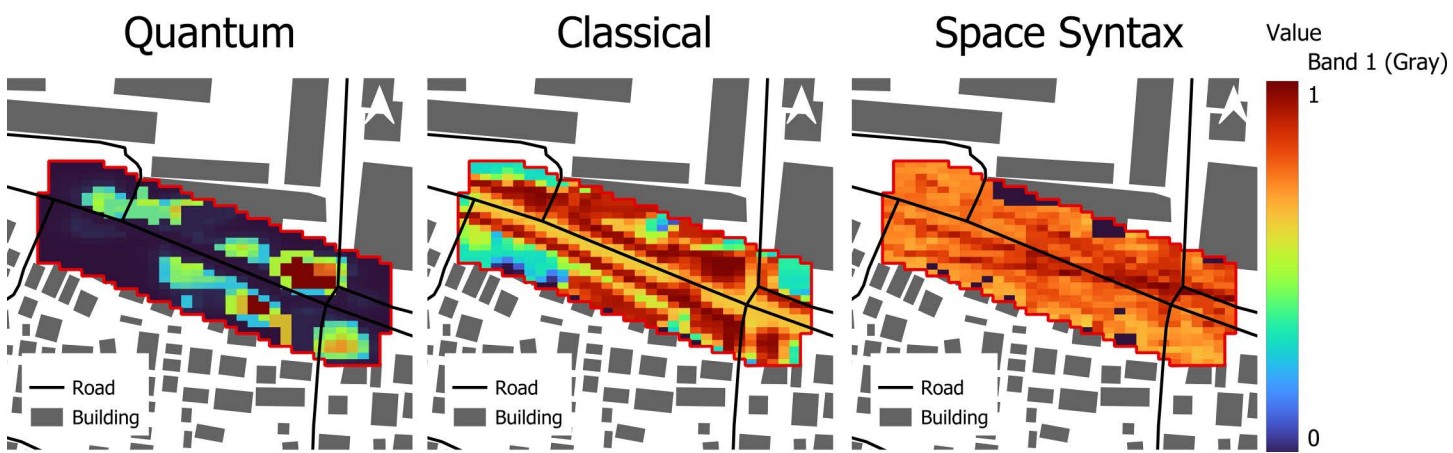

**Fig 10. Comparative panels (Observed vs. Quantum vs. Classical vs. Syntax) for evening period.** (Source: Prepared by authors).

**Table 7. Temporal and aggregate performance of the quantum-inspired model and bootstrap uncertainty.**

**(a) Period-wise performance (link-level pedestrian shares)**

| Period | n | Pearson r | Spearman ρ | RMSE |
|---|---|---|---|---|
| Morning | 568 | 0.76573 | 0.742 | 39,224.81 |
| Midday | 569 | 0.70874 | 0.68964 | 44,152.64 |
| Evening | 568 | 0.76718 | 0.74794 | 38,643.09 |

**(b) Aggregate performance with bootstrap confidence intervals (B = 1000)**

| Metric | Point estimate | 95% CI low | 95% CI high |
|---|---|---|---|
| Pearson r | 0.65711 | 0.61289 | 0.69914 |
| Spearman ρ | 0.64026 | 0.60956 | 0.66941 |

Note: Refer S3 Table in Supplement materials.

**Aggregate performance and bootstrap uncertainty.** To complement period-wise metrics, we pooled observations across all periods and applied nonparametric bootstrap resampling (B = 1000). The resulting point estimates indicate a moderate-to-strong global association between observed and predicted pedestrian shares, with Pearson's r = 0.657 and Spearman's ρ = 0.640 (Table 5). Bootstrap-based 95% confidence intervals quantify uncertainty: r lies between 0.613 and 0.699, and ρ between 0.610 and 0.669. These intervals are relatively narrow and remain well above zero, indicating that the observed predictive skill is robust and statistically meaningful rather than an artefact of a particular period or sample. The fact that the pooled estimates are lower than the best period-specific correlations is expected and reflects underlying temporal heterogeneity between morning, midday, and evening conditions.

**Cross-sectional robustness (k-fold analysis).** We further examined spatial robustness using a five-fold partitioning of grid cells as pseudo-test sets (S3 Table). Across folds, Pearson's r varies between 0.617 and 0.709 (mean 0.659, SD 0.036), and Spearman's ρ between 0.567 and 0.696 (mean 0.638, SD 0.054), with RMSE around 34,556 (SD ≈ 3,266). No fold exhibits a collapse in performance, indicating that the model's predictive capability is not driven by a small set of particularly favorable locations. The relatively low standard deviations across folds suggest that the quantum-inspired formulation generalizes well across spatial subsets with differing combinations of urban form, shade, and crowding.

**Implications for a quantum-inspired probability field.** Taken together, the temporal, bootstrap, and k-fold analyses provide converging evidence of robustness: the model maintains stable performance across diurnal periods, across resampled datasets, and across spatial partitions. Unlike classical machine-learning models that are retrained to reproduce a catalogue of scenarios, the quantum-inspired framework calibrates a potential field once and then updates scenarios by modifying attraction/repulsion components (e.g., barriers, events, shade changes) without retraining. The probability distribution arises from the eigen-solution of this field rather than from deterministic route assignment. The uncertainty analyses reported here therefore quantify the reliability of the calibrated probability field under observed conditions, while the scenario experiments in Scenario responsive section illustrate how the same quantum-inspired representation can be logically adapted to new operational situations.

## Conclusion and recommendations

### Summary of key empirical findings

This study developed and tested a quantum-inspired probabilistic framework for pedestrian mobility at a compact, heat-exposed junction in Moratuwa, Sri Lanka. Pedestrian presence was represented as a spatial probability field shaped by a composite potential that combines built form, microclimate (shade/ solar exposure), and time-varying crowd density. Calibrated against link-level pedestrian shares, the model achieved Pearson correlations of approximately 0.77 in the morning and evening and 0.71 at midday, with corresponding Spearman correlations around 0.69–0.75. When pooling all periods

and applying nonparametric bootstrap resampling (B = 1000), the global performance remained robust, with r = 0.657 (95% CI 0.613–0.699) and ρ = 0.640 (95% CI 0.610–0.669). A five-fold spatial stability analysis yielded a mean r of 0.659 (SD 0.036) and mean ρ of 0.638 (SD 0.054), indicating that results are not driven by a single favorable subset of links. Scenario analyses showed that once the potential field is calibrated, probability distributions adapt logically to temporary barriers and events without retraining, producing re-routing patterns consistent with field observations. Comparisons with a classical Boltzmann-type potential and a space-syntax predictor further demonstrated higher linear and rank agreement with observations and more realistic spatial clustering for the quantum-inspired model.

## Practical implications for design and operations

Within this junction-scale, tropical setting, the model consistently highlights shade and microclimate as primary drivers of pedestrian redistribution. At midday, probability mass shifts into shaded, well-connected corridors and away from exposed verges, even when geometry alone would favor the latter. In the evening, as thermal stress diminishes, highest probabilities emerge along well-connected exits and edges adjacent to points of interest, still modulated by available shade. These patterns provide an operational rule of thumb for comparable contexts: where resources are limited, interventions that increase shade along already connected pedestrian routes—through trees, arcades, or light structures—are likely to yield stronger gains in pedestrian presence than simply adding capacity on exposed segments.

Because outputs are normalized probability fields defined on a regular grid and link network, they can be converted into expected changes in link-level pedestrian shares associated with marginal changes in shade, barriers, or connectivity. This makes the framework suitable for ranking candidate interventions (e.g., where to plant trees first, which informal parking areas to restrict, where to formalize crossings) in terms of their expected impact on pedestrian presence. The scenario experiments illustrate how day-to-day operations can also be tested virtually: modifying the potential to reflect an event perimeter, parked vehicles, or a protest footprint immediately yields new probability maps without refitting the model, providing a fast way to assess possible crowding or conflict hotspots before making on-the-ground changes.

## Theoretical and methodological contributions

Theoretically, the study demonstrates how pedestrian movement at junction scale can be reframed as a ground-state probability field in a composite potential, rather than as a set of deterministic routes or a purely black-box prediction. The decomposition into a structural component $V_{urbs}$ and a dynamic component $V_{civitas}$ provides a clear way to encode edges, crossings, barriers, visibility and points of interest alongside shade and crowding, and to interpret how these ingredients jointly form low-effort corridors and repulsive zones. The "quantum-inspired" aspect lies in the use of superposition and eigen-solutions as tools to represent multi-route choice and spatial regularization; it does not imply any literal quantum physical processes in urban movement.

Methodologically, the work offers a reproducible workflow based on open tools and standard data structures. All inputs are first rasterized to a 5 m × 5 m grid and 15-minute temporal index aligned with the counting protocol, then down sampled to a 32 × 32 lattice compatible with near-term quantum-device constraints while retaining key morphological and microclimatic gradients. The model integrates configuration, microclimate, and behavior in a single potential, learns weights that can be interpreted through a proxy-based multicollinearity treatment, and outputs GeoTIFF probability fields and link-level time series. In comparative terms, the quantum-inspired model outperforms classical Boltzmann-type potential and a space-syntax predictor in both linear and rank agreement. Moran's I is interpreted conservatively as a relative indicator of clustering, recognizing that the smoothness term inherently increases spatial autocorrelation.

## Limitations and scope of applicability

Several limitations and boundaries of applicability need to be stressed. First, the model is calibrated and validated at a single junction in a hot, humid environment where shade and thermal comfort are particularly salient. In cooler or high-latitude cities, or in seasons when solar exposure is desirable, the weight and direction of shade-related effects may

change substantially. The present parameter values should therefore be understood as context-specific, providing evidence for similar tropical or subtropical street conditions rather than universally transferable coefficients.

Second, the framework assumes a largely continuous, surface-level pedestrian domain with conventional at-grade crossings. It does not explicitly represent multi-level stations, grade-separated crossings, or extensive indoor networks. In highly layered environments, a two-dimensional potential may under-represent vertical constraints and route options, and extensions to multi-layer or 3D formulations would be required. Third, traffic and signal control are treated in a simplified way. Vehicle density is included as a contextual factor, but detailed signal timing, pedestrian phases, and turning rules are not modelled explicitly. In heavily signalized networks where such elements dominate delay and route choice, predictive performance may be weaker and the model should be complemented by more detailed traffic-control representations.

Fourth, dynamic drivers in the potential are deliberately restricted to crowd density and shade or solar exposure, reflecting both model parsimony and current hybrid/quantum optimization limits. Other time-varying influences—such as short-lived curb uses, detailed retail spillover, or high-frequency meteorological variation—are approximated using static proxies or scenario flags. Shadow fields rely on geometry and period aggregates rather than continuous microclimate measurements, so thermal conditions are only approximated. Finally, calibration is based on fixed-point counts mapped to links and grid cells, supported by photos and short video clips, rather than full individual trajectories. As a result, the model reproduces link-level presence rather than path-level choice. In contexts where within-link heterogeneity, crossing paths, or microscopic conflict dynamics are central, trajectory-based data and models may be more suitable. The inclusion of Laplacian and light Gaussian smoothing, necessary for numerical stability and corridor coherence, also attenuates very small-scale anomalies, which may cause the model to understate highly localized peaks or troughs.

Taken together, these considerations suggest that the current formulation is most appropriate for junction-scale, surface-level environments in warm climates, where shade and crowding are known or expected to be strong behavioral drivers and where at least coarse microclimate and count data are available.

## Directions for future research

Future work should first test cross-city and cross-climate transferability by applying the framework to multiple junctions differing in morphology, culture and climate, and by exploring hierarchical calibration that separates broadly consistent effects from site-specific ones. Closer coupling with urban microclimate and traffic-signal models could provide time-continuous thermal fields and more realistic representations of delays and crossing opportunities, particularly in more complex road networks. Combining the present link-based calibration with trajectories derived from GPS, Bluetooth, Wi-Fi or computer-vision systems would permit validation at the path scale, support modelling of within-link heterogeneity, and enable exploration of more explicitly "quantum-like" interference effects in route choice.

Extensions toward multimodal and equity-focused analyses are also promising. By adapting the potential to include accessibility for people with limited mobility, interactions with cyclists, or perceived safety signals, the same probability-field approach could help reveal who gains or losses from different shading or connectivity strategies. Finally, embedding the probability fields in multi-objective planning and decision-support tools would allow explicit trade-offs between comfort, safety, travel time, and implementation cost under uncertainty, making the method more directly usable in policy and design processes.

## Final remarks

In conclusion, this study shows that a probabilistic, quantum-inspired formulation can provide spatially explicit and empirically grounded insight into how pedestrians redistribute under changing structural, climatic, and crowding conditions, at least for compact tropical junctions. By casting movement as a probability field shaped by a composite, interpretable potential rather than as a set of fixed routes, the approach unifies configuration, microclimate and behavior in a single, reproducible framework and improves empirical fit relative to the classical and space-syntax benchmarks examined here.

At the same time, it remains subject to clear contextual and data-related limitations, and its assumptions will not hold in all urban settings. The main contribution is therefore not a universally applicable tool, but a demonstrated proof-of-concept for using quantum-inspired probability fields in real pedestrian environments, together with a set of methodological and empirical insights that can guide adaptations to other cities, climates and network complexities.

## Supporting information

**S1 Table. Pearson correlation matrix of built-form, movement, and microclimatic covariates.**
(DOCX)

**S2 Table. Variance inflation factors (VIFs) for spatial covariates.**
(DOCX)

**S3 Table. Five-fold spatial stability analysis for the quantum-inspired model.**
(DOCX)

## Acknowledgments

The authors gratefully acknowledge the University of Moratuwa, Sri Lanka, for providing open-access publication support through the Senate Research Committee Conference & Publishing Support Grant.

During the preparation of this work, the author(s) used OpenAI ChatGPT to improve grammar, clarity, and structure. The author(s) reviewed and edited all content and take full responsibility for the manuscript's integrity. No AI was used for data analysis, results generation, figures/tables from data, or reference creation.

## Author contributions

**Conceptualization:** Malith Deshan, Amila Jayasinghe, Chethika Abenayake.

**Data curation:** Malith Deshan.

**Formal analysis:** Malith Deshan.

**Funding acquisition:** Amila Jayasinghe.

**Investigation:** Malith Deshan, Amila Jayasinghe.

**Methodology:** Malith Deshan, Amila Jayasinghe.

**Project administration:** Amila Jayasinghe.

**Resources:** Malith Deshan, Amila Jayasinghe.

**Software:** Malith Deshan.

**Supervision:** Amila Jayasinghe.

**Validation:** Malith Deshan, Amila Jayasinghe.

**Visualization:** Malith Deshan.

**Writing – original draft:** Amila Jayasinghe, Chethika Abenayake.

**Writing – review & editing:** Malith Deshan, Amila Jayasinghe, Chethika Abenayake.

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
