## [Decision Letter · Decision Letter 0]

7 Jan 2026

PONE-D-25-62891Quantum-Inspired Pedestrian Mobility Modeling: Applying Probabilistic Spatial Simulation to Urban Walkability and Thermal Comfort in Sri LankaPLOS One

Dear Dr. Jayasinghe,

Thank you for submitting your manuscript to PLOS ONE. After careful consideration, we feel that it has merit but does not fully meet PLOS ONE’s publication criteria as it currently stands. Therefore, we invite you to submit a revised version of the manuscript that addresses the points raised during the review process.

**When revising your manuscript, please consider all issues mentioned in the reviewers' comments carefully: please outline every change made in response to their comments and provide suitable rebuttals for any comments not addressed. Please note that your revised submission may need to be re-reviewed.**

We look forward to receiving your revised manuscript.

Kind regards,

Genyu Xu, Ph.D.

Academic Editor

PLOS One

**Journal Requirements:**

1. When submitting your revision, we need you to address these additional requirements. Please ensure that your manuscript meets PLOS ONE's style requirements, including those for file naming. The PLOS ONE style templates can be found at https://journals.plos.org/plosone/s/file?id=wjVg/PLOSOne_formatting_sample_main_body.pdf and https://journals.plos.org/plosone/s/file?id=ba62/PLOSOne_formatting_sample_title_authors_affiliations.pdf 2. Please note that PLOS One has specific guidelines on code sharing for submissions in which author-generated code underpins the findings in the manuscript. In these cases, we expect all author-generated code to be made available without restrictions upon publication of the work. Please review our guidelines at https://journals.plos.org/plosone/s/materials-and-software-sharing#loc-sharing-code and ensure that your code is shared in a way that follows best practice and facilitates reproducibility and reuse. 3. We note that the grant information you provided in the ‘Funding Information’ and ‘Financial Disclosure’ sections do not match.  When you resubmit, please ensure that you provide the correct grant numbers for the awards you received for your study in the ‘Funding Information’ section. 4. In the online submission form you indicate that your data is not available for proprietary reasons and have provided a contact point for accessing this data. Please note that your current contact point is a co-author on this manuscript. According to our Data Policy, the contact point must not be an author on the manuscript and must be an institutional contact, ideally not an individual. Please revise your data statement to a non-author institutional point of contact, such as a data access or ethics committee, and send this to us via return email. Please also include contact information for the third party organization, and please include the full citation of where the data can be found. 5. When completing the data availability statement of the submission form, you indicated that you will make your data available on acceptance. We strongly recommend all authors decide on a data sharing plan before acceptance, as the process can be lengthy and hold up publication timelines. Please note that, though access restrictions are acceptable now, your entire data will need to be made freely accessible if your manuscript is accepted for publication. This policy applies to all data except where public deposition would breach compliance with the protocol approved by your research ethics board. If you are unable to adhere to our open data policy, please kindly revise your statement to explain your reasoning and we will seek the editor's input on an exemption. Please be assured that, once you have provided your new statement, the assessment of your exemption will not hold up the peer review process. 6. We note that Figures 1, 2, 5, 6, 7, 9 and 10 in your submission contain map/satellite images which may be copyrighted. All PLOS content is published under the Creative Commons Attribution License (CC BY 4.0), which means that the manuscript, images, and Supporting Information files will be freely available online, and any third party is permitted to access, download, copy, distribute, and use these materials in any way, even commercially, with proper attribution. For these reasons, we cannot publish previously copyrighted maps or satellite images created using proprietary data, such as Google software (Google Maps, Street View, and Earth). For more information, see our copyright guidelines: http://journals.plos.org/plosone/s/licenses-and-copyright. We require you to either present written permission from the copyright holder to publish these figures specifically under the CC BY 4.0 license, or remove the figures from your submission: a. You may seek permission from the original copyright holder of Figures 1, 2, 5, 6, 7, 9 and 10 to publish the content specifically under the CC BY 4.0 license.   We recommend that you contact the original copyright holder with the Content Permission Form (http://journals.plos.org/plosone/s/file?id=7c09/content-permission-form.pdf) and the following text:“I request permission for the open-access journal PLOS ONE to publish XXX under the Creative Commons Attribution License (CCAL) CC BY 4.0 (http://creativecommons.org/licenses/by/4.0/). Please be aware that this license allows unrestricted use and distribution, even commercially, by third parties. Please reply and provide explicit written permission to publish XXX under a CC BY license and complete the attached form.” Please upload the completed Content Permission Form or other proof of granted permissions as an "Other" file with your submission. In the figure caption of the copyrighted figure, please include the following text: “Reprinted from [ref] under a CC BY license, with permission from [name of publisher], original copyright [original copyright year].” b. If you are unable to obtain permission from the original copyright holder to publish these figures under the CC BY 4.0 license or if the copyright holder’s requirements are incompatible with the CC BY 4.0 license, please either i) remove the figure or ii) supply a replacement figure that complies with the CC BY 4.0 license. Please check copyright information on all replacement figures and update the figure caption with source information. If applicable, please specify in the figure caption text when a figure is similar but not identical to the original image and is therefore for illustrative purposes only.The following resources for replacing copyrighted map figures may be helpful: USGS National Map Viewer (public domain): http://viewer.nationalmap.gov/viewer/The Gateway to Astronaut Photography of Earth (public domain): http://eol.jsc.nasa.gov/sseop/clickmap/Maps at the CIA (public domain): https://www.cia.gov/library/publications/the-world-factbook/index.html and https://www.cia.gov/library/publications/cia-maps-publications/index.htmlNASA Earth Observatory (public domain): http://earthobservatory.nasa.gov/Landsat:
http://landsat.visibleearth.nasa.gov/USGS EROS (Earth Resources Observatory and Science (EROS) Center) (public domain): http://eros.usgs.gov/#Natural Earth (public domain): http://www.naturalearthdata.com/ 7. Please upload a new copy of Figures 4 and 8, as the detail is not clear. Please follow the link for more information:  https://journals.plos.org/plosone/s/figures 8. If the reviewer comments include a recommendation to cite specific previously published works, please review and evaluate these publications to determine whether they are relevant and should be cited. There is no requirement to cite these works unless the editor has indicated otherwise.

Reviewers' comments:

Reviewer's Responses to Questions

**Comments to the Author**

1. Is the manuscript technically sound, and do the data support the conclusions?

Reviewer #1: Yes

Reviewer #2: Partly

2. Has the statistical analysis been performed appropriately and rigorously? 

Reviewer #1: Yes

Reviewer #2: Yes

3. Have the authors made all data underlying the findings in their manuscript fully available?

Reviewer #1: No

Reviewer #2: No

4. Is the manuscript presented in an intelligible fashion and written in standard English?

Reviewer #1: Yes

Reviewer #2: Yes

5. Review Comments to the Author

**Reviewer #1:** The topic of quantum inspired modeling of pedestrian mobility is very interesting. However, the quality of the presentation needs great improvement. Here is more detailed comments:

1. In the introduction, Line 92, "To address this need"... To address what needs? The gap is not clear. The authors provide a concise summary of previous pedestrian modelling approach, but, can quantum modelling solve the limitations of previous techniques? Please provide a clearer gap.

2. Yes, the study proposed a novel and appealing approach, but following the previous point, what is the research contribution in comparison with previous methods? Greater accuracy? Or Better explainability? Or others?

3. I found the quality of many figures are very low, with low resolution and too small fonts, please comprehensively improve.

4. Figure 1, what is the use of the 500m buffer?

5. In the data collection, counts of pedestrian is not enough. The spatial location, or even trajectory of pedestrians are also important. Did the study obtain more fine grained feature of the pedestrian?

6. An overall workflow figure of how the modeling and validation were conducted would help with the clarity.

7. Figure 5, the title is confusing, with unrepresented codes, and name of vertical axis (V urbs V Civitas V Total) is not self-explainable.

8. Too litter references. The study should provide a much more comprehensive literature.

Overall, the authors should work much hard to increase the overall theory depth and presentation quality. Otherwise, it is a waste of such an interesting idea.

**Reviewer #2:** This study introduces a quantum-inspired framework to investigate pedestrian mobility under uncertainty, with particular attention to shade, thermal comfort, and crowding. The idea is interesting; however, in order to meet the standard for publication, the manuscript requires the following revisions.

1. In the Introduction, the concept of “quantum-inspired” is mainly presented at an analogical level, and its essential differences from conventional probabilistic models or random-field–based approaches are not clearly articulated. It is recommended that the authors explicitly clarify the non-substitutable aspects of this approach in terms of modeling assumptions or solution mechanisms.

2. The manuscript treats shade and thermal comfort as key drivers of changes in pedestrian probability distributions. However, the Introduction provides only a limited review of existing studies on pedestrian thermal comfort, heat exposure, or microclimate effects on route choice, making it unclear how the present study advances this line of research. Additional relevant literature review is recommended.

3. Spatial and temporal discretization scales play a decisive role in shaping the resulting probability field. In the Methods section, the choice of a 5 m × 5 m grid and a 15-minute temporal resolution lacks empirical justification or sensitivity analysis. It is recommended to provide clearer justification and relevant literature support for these parameter settings.

4. The model treats variables such as building height, building density, and shade intensity as relatively independent inputs. In practice, these variables are often highly correlated and partially causally coupled. Screening by simple correlation coefficients alone may be insufficient to address structural multicollinearity. A more systematic treatment of variable interdependence is recommended, together with a clearer explanation of which variables are used as proxies.

5. Model performance is mainly evaluated using Pearson’s r, Spearman’s ρ, and Moran’s I. However, confidence intervals, significance testing, or stability analysis across different days or time periods are not reported, which may lead to an overestimation of model performance. It is recommended to include uncertainty quantification and report plausible ranges of performance variation.

6. The reported Moran’s I values require cautious interpretation. The model explicitly incorporates spatial regularization and smoothing procedures, which may themselves enhance spatial autocorrelation. Under the current design, it is difficult to distinguish whether the predicted spatial clustering arises from pedestrian behavior or from the model structure. Comparative analyses that weaken or remove smoothing terms are recommended, or alternatively, the role of Moran’s I as evidence of behavioral realism should be interpreted more conservatively.

7. Although several limitations are acknowledged, the Conclusion still places strong emphasis on the advantages of the proposed method. Potential conditions under which the model may perform poorly—such as application to other cities, different climatic contexts, or more complex traffic environments—are not sufficiently discussed. It is recommended to more clearly delineate the scope and applicability boundaries of the model.

8. The Conclusion repeatedly aligns the study with the journal’s mission, but the linkage to specific empirical results is not always explicit. It is recommended to reduce submission-oriented statements and instead allow concrete quantitative findings and methodological characteristics to naturally address the journal’s thematic focus.

9. In Section 5, the term “Applied Geography” is capitalized. Please clarify why capitalization is used here.

6. PLOS authors have the option to publish the peer review history of their article (what does this mean?). If published, this will include your full peer review and any attached files.

Reviewer #1: No

Reviewer #2: No

---

## [Author Response · Author response to Decision Letter 1]

5 Mar 2026

Response to Reviewers

Dear Editor,

We would like to thank the Journal of PLOS ONE for giving us the opportunity to revise

Manuscript PONE-D-25-62891

Quantum-Inspired Pedestrian Mobility Modeling: Applying Probabilistic Spatial Simulation to Urban Walkability and Thermal Comfort in Sri Lanka.

We thank the reviewers for their constructive comments. We have carefully taken their comments into consideration in preparing our revision. Below is our response to their comments.

Thanks for all comments.

Best wishes,

Amila Jayasinghe

Response to Editor/Journal Requirements

1. Please ensure that your manuscript meets PLOS ONE's style requirements, including those for file naming..

Response: We have rearranged the manuscript according to the PLOS ONE's style requirements.

2. Please note that PLOS One has specific guidelines on code sharing for submissions in which author-generated code underpins the findings in the manuscript. In these cases, we expect all author-generated code to be made available without restrictions upon publication of the work. Please review our guidelines at https://journals.plos.org/plosone/s/materials-and-software-sharing#loc-sharing-code and ensure that your code is shared in a way that follows best practice and facilitates reproducibility and reuse

In the online submission form you indicate that your data is not available for proprietary reasons and have provided a contact point for accessing this data. Please note that your current contact point is a co-author on this manuscript. According to our Data Policy, the contact point must not be an author on the manuscript and must be an institutional contact, ideally not an individual. Please revise your data statement to a non-author institutional point of contact, such as a data access or ethics committee, and send this to us via return email. Please also include contact information for the third party organization, and please include the full citation of where the data can be found.

When completing the data availability statement of the submission form, you indicated that you will make your data available on acceptance. We strongly recommend all authors decide on a data sharing plan before acceptance, as the process can be lengthy and hold up publication timelines. Please note that, though access restrictions are acceptable now, your entire data will need to be made freely accessible if your manuscript is accepted for publication. This policy applies to all data except where public deposition would breach compliance with the protocol approved by your research ethics board. If you are unable to adhere to our open data policy, please kindly revise your statement to explain your reasoning and we will seek the editor's input on an exemption. Please be assured that, once you have provided your new statement, the assessment of your exemption will not hold up the peer review process.

Response: Thank you for your message and for the guidance regarding code and data sharing. We have now updated our materials to comply with the PLOS ONE policies.

1. Code sharing

All author-generated code that underpins the analyses and figures in the manuscript has been deposited in an open repository on GitHub:

https://github.com/malithdeshan1222/PedestrianQuantumSimulation

We will ensure that this link is cited in the manuscript’s Data and Code Availability section and that the repository remains openly accessible upon publication.

2. Data availability and institutional contact

We acknowledge that our original Data Availability Statement listed a co-author as the contact point, which does not conform to PLOS ONE’s policy. We have now revised the statement to provide a non-author, institutional contact for access to the third-party data.

Response: We confirm that this study did not receive support from a specific research grant with a grant number. Part of the article processing charge will be supported by the University of Moratuwa under its internal scheme for “Conference Attendance and Open-Access Publishing.” We have updated the Funding Information and Financial Disclosure sections to reflect this consistently.

Response:

5. We note that Figures 1, 2, 5, 6, 7, 9 and 10 in your submission contain map/satellite images which may be copyrighted. All PLOS content is published under the Creative Commons Attribution License (CC BY 4.0), which means that the manuscript, images, and Supporting Information files will be freely available online, and any third party is permitted to access, download, copy, distribute, and use these materials in any way, even commercially, with proper attribution. For these reasons, we cannot publish previously copyrighted maps or satellite images created using proprietary data, such as Google software (Google Maps, Street View, and Earth). For more information, see our copyright guidelines: http://journals.plos.org/plosone/s/licenses-and-copyright.

We require you to either present written permission from the copyright holder to publish these figures specifically under the CC BY 4.0 license, or remove the figures from your submission:

a. You may seek permission from the original copyright holder of Figures 1, 2, 5, 6, 7, 9 and 10 to publish the content specifically under the CC BY 4.0 license.

Response: Thank you for drawing our attention to this issue. In the revised submission we have removed all figures that relied on Google-based or otherwise non-CC BY 4.0–compatible imagery and replaced them with maps derived from open or self-generated sources.

Specifically, Figures 1 and 2 have been redrawn using our own GIS layers and OpenStreetMap data as a basemap. The figures and captions now explicitly acknowledge the source as “© OpenStreetMap contributors” and comply with the ODbL/CC-BY-compatible licensing terms. Figures 5, 6, 7, and 9 have been revised to remove any Google imagery; they now use only our own survey data, schematic representations, and/or OSM-derived base layers. No Google Maps, Google Earth, Street View, or other proprietary satellite or map images remain in the revised manuscript or Supporting Information.

We trust that the updated figures now fully comply with PLOS ONE’s copyright and CC BY 4.0 requirements.

7. Please upload a new copy of Figures 4 and 8, as the detail is not clear. Please follow the link for more information: https://journals.plos.org/plosone/s/figures

Response: In the revised submission we have uploaded new high-resolution versions of Figures 4 and 8.

Response to Reviewers' comments:

Reviewer #1

The topic of quantum inspired modeling of pedestrian mobility is very interesting. However, the quality of the presentation needs great improvement. Here is more detailed comments:

Comment 1: In the introduction, Line 92, "To address this need"... To address what needs? The gap is not clear. The authors provide a concise summary of previous pedestrian modelling approach, but, can quantum modelling solve the limitations of previous techniques? Please provide a clearer gap.

Response 1

Thank you for pointing out that the research gap was not clearly articulated. In the revised Introduction, we now state explicitly what is missing in existing approaches and how our quantum-inspired model is designed to address it.

We added the following sentence to clarify the gap created by conventional pedestrian models:

“Collectively, these approaches struggle to represent two features that are central to contemporary street-level planning: (i) non-deterministic, multi-route choice under uncertainty and (ii) the way microclimate, shade, and crowding reconfigure pedestrian patterns over the course of a day.” (Pg 2 Lines 45-48)

Comment 2: Yes, the study proposed a novel and appealing approach, but following the previous point, what is the research contribution in comparison with previous methods? Greater accuracy? Or Better explainability? Or others?

Response 2

We agree that the contributions relative to existing methods needed to be more explicit. In the revised Introduction, we added a dedicated paragraph that lists four concrete contributions. For example, we now write:

“Within this framing, our study makes four contributions to urban planning and design and to the broader field of urban informatics. First, we develop a quantum-inspired, eigen-based pedestrian model that is explicitly grounded in urban design variables—connectivity, crossings, barriers, visibility, points of interest—and in microclimate and crowding, rather than in abstract network states. Second, we systematically integrate thermal comfort and shade into the movement model, treating them as first-order components of the potential field and demonstrating how they reconfigure probability distributions across the day in a hot tropical context. Third, we benchmark the quantum-inspired model against a classical Boltzmann-type potential model and a space-syntax-based predictor, showing gains in statistical fit and, importantly, more realistic spatial patterns that planners can interpret and act upon. Fourth, we embed the method in an urban informatics workflow … so that it can be transferred to data-constrained settings typical of many cities in the Global South.” (Pg 4 Lines 94-105)

This new paragraph explicitly clarifies that our contributions are both quantitative (improved fit relative to classical baselines) and qualitative/interpretive (probability fields that are more interpretable and useful for urban planning and design).

Comment 3: I found the quality of many figures are very low, with low resolution and too small fonts, please comprehensively improve.

Response 3:

Thank you for pointing this out. In the revised submission we have uploaded new high-resolution versions of Figures (all Updated Figures 1–10). The updated figures use increased font sizes, clearer legends, and higher DPI export settings so that all labels, symbols, and map details are legible in both on-screen and print formats.

Comment 4: Figure 1, what is the use of the 500m buffer?

Response 4:

We appreciate this observation. In the original submission, the 500 m buffer shown in Figure 1 was intended only as a visual context window around the junction, not as an analytical boundary. We agree that this was potentially confusing. In the revised manuscript we have removed the 500 m buffer from Figure 1, and the figure now displays only the actual study area used for modelling and validation. The Methods section also clarifies that all analyses are confined to the delineated junction grid, without using a 500 m buffer as a modelling unit. (“Study area” subsection; revised Figure 1 in Pg 6)

Comment 5: In the data collection, counts of pedestrian is not enough. The spatial location, or even trajectory of pedestrians are also important. Did the study obtain more fine grained feature of the pedestrian?

Response 5:

We agree that spatial location and trajectories provide valuable additional information. In the revised manuscript we clarify that, in addition to synchronized fixed-point counts, observers captured photos and short video clips during each survey session. These materials were used to verify and refine the mapping of pedestrians to specific links, sides, and crossings, so that the calibration targets represent link-level pedestrian presence rather than undifferentiated totals. Individual trajectories were not extracted, and we now explicitly acknowledge this as a limitation while identifying the integration of GPS-/video-based trajectories as a key avenue for future work (Pg 8 Lines 160-162)

Comment 6: An overall workflow figure of how the modelling and validation were conducted would help with the clarity.

Response 6:

Thank you for this helpful suggestion. In the revised manuscript we have added a new Figure 3, which presents an overall workflow of the study. (Pg 12 Fig 3)

Comment 7: Figure 5, the title is confusing, with unrepresented codes, and name of vertical axis (V urbs V Civitas V Total) is not self-explainable.

Response 7:

We appreciate this comment. In the revised manuscript we have clarified both the title and the labelling of Figure 5. The figure is now titled “Static, dynamic, and total potential fields by time of day,” and the vertical labels have been changed from the abbreviated codes to self-explanatory terms: “Static potential (V_urbs)”, “Dynamic potential (V_civitas)”, and “Total potential (V_total)”. The caption explicitly explains that V_urbs represents the structural (built form) component, V_civitas the dynamic behavioral component, and that blue corresponds to low/attractive potential while red corresponds to high/repulsive potential. We hope this makes the figure much clearer to readers unfamiliar with the notation. (Pg 22 Fig 5)

Comment 8: Too litter references. The study should provide a much more comprehensive literature.

Response 8:

We have substantially expanded the literature review in the Introduction in two directions:

1. Broader coverage of pedestrian modelling traditions: We now explicitly discuss gravity/entropy models, space syntax and network centrality, agent-based models, and deep learning approaches, including their strengths and limitations, e.g.:

“A wide range of modeling traditions has contributed to understanding pedestrian flows. Gravity and entropy-based models capture aggregate flows… Space syntax and related network-centrality approaches emphasize the role of configurational accessibility… Agent-based models and, more recently, deep learning frameworks can reproduce complex trajectories from data, but they are typically data-hungry, computationally intensive, and often opaque to planners…”

2. Expanded coverage of thermal comfort and microclimate in pedestrian behaviour: In response to both Reviewer #1 and Reviewer #2, we added a dedicated paragraph summarizing empirical work on heat, shade, outdoor comfort, and microclimate effects on walking:

“Recent work has begun to document how thermal comfort and microclimate systematically influence pedestrian behavior. Empirical and experimental studies show that pedestrians in hot climates adjust their routes to maximize shade, discounting sun-exposed distance and sometimes preferring longer shaded paths… Urban street-canyon studies link street geometry, sky-view factor, and material properties to thermal indices such as PET or UTCI… Microclimate mapping and walk-along surveys further demonstrate that local shade, radiation, and wind drive perceived comfort, walking experience, and sidewalk usage…” (Pg 22 Line 49 onwards till Pg Line 65)

We also introduced additional references on urban informatics, quantum-inspired decision modelling, and quantum/quantum-inspired mobility models, situating our work more firmly at the intersection of urban science, computer science, and geoinformatics.

Reviewer #2

This study introduces a quantum-inspired framework to investigate pedestrian mobility under uncertainty, with particular attention to shade, thermal comfort, and crowding. The idea is interesting; however, in order to meet the standard for publication, the manuscript requires the following revisions.

Comment 1: In the Introduction, the concept of “quantum-inspired” is mainly presented at an analogical level, and its essential differences from conventional probabilistic models or random-field–based approaches are not clearly articulated. It is recommended that the authors explicitly clarify the non-substitutable aspects of this approach in terms of modeling assumptions or solution mechanisms.

Response 1:

We appreciate this important comment. In the revised Introduction, we now go beyond analogy and explicitly state the non-substitutable modelling assumptions and mechanisms that distinguish our quantum-inspired approach from standard probabilistic models.

We added the following conceptual description:

“Conceptually, we treat pedestrians not as committing instantaneously to a single ‘shortest’ path, but as occupying a superposed choice state over multiple feasibl

---

## [Decision Letter · Decision Letter 1]

19 Apr 2026

Quantum-Inspired Pedestrian Mobility Modeling: Applying Probabilistic Spatial Simulation to Urban Walkability and Thermal Comfort in Sri Lanka

PONE-D-25-62891R1

Dear Dr. Jayasinghe,

We’re pleased to inform you that your manuscript has been judged scientifically suitable for publication and will be formally accepted for publication once it meets all outstanding technical requirements.

Kind regards,

Genyu Xu, Ph.D.

Academic Editor

PLOS One

Additional Editor Comments (optional):

Reviewers' comments:

Reviewer's Responses to Questions

**Comments to the Author**

1. If the authors have adequately addressed your comments raised in a previous round of review and you feel that this manuscript is now acceptable for publication, you may indicate that here to bypass the “Comments to the Author” section, enter your conflict of interest statement in the “Confidential to Editor” section, and submit your "Accept" recommendation.

Reviewer #1: All comments have been addressed

Reviewer #2: All comments have been addressed

2. Is the manuscript technically sound, and do the data support the conclusions?

Reviewer #1: Yes

Reviewer #2: Yes

3. Has the statistical analysis been performed appropriately and rigorously? 

Reviewer #1: Yes

Reviewer #2: Yes

4. Have the authors made all data underlying the findings in their manuscript fully available?

Reviewer #1: No

Reviewer #2: Yes

5. Is the manuscript presented in an intelligible fashion and written in standard English?

Reviewer #1: Yes

Reviewer #2: Yes

6. Review Comments to the Author

Reviewer #1: The authors have sufficiently addressed my comments and I agree that the paper can now be published.

Reviewer #2: All of my previous comments have been adequately addressed in the revised version, and no major concerns remain at this stage.

7. PLOS authors have the option to publish the peer review history of their article (what does this mean?). If published, this will include your full peer review and any attached files.

Reviewer #1: No

Reviewer #2: No

---

## [Editor Report · Acceptance letter]

PONE-D-25-62891R1

PLOS One

Dear Dr. Jayasinghe,

I'm pleased to inform you that your manuscript has been deemed suitable for publication in PLOS One. Congratulations! Your manuscript is now being handed over to our production team.

Kind regards,

on behalf of

Dr. Genyu Xu

Academic Editor

PLOS One